# DAA: Amplifying Unknown Discrepancy for Test-Time Discovery

**Tianle Liu[1,2], Fan Lyu[*3], Chenggong Ni[1], Zhang Zhang[3], Fuyuan Hu[* 1,4,5], Liang Wang[3]**

[1]School of Electronics and Information Engineering, Suzhou University of Science and Technology
[2]Suzhou Key Laboratory of Embodied Intelligent Agents
for Cooperative Perception and Advanced Control
[3]New Laboratory of Pattern Recognition, Institute of Automation, Chinese Academy of Sciences
[4]Jiangsu Industrial Intelligent and Low-carbon Technology Engineering Center
[5]Suzhou Key Laboratory of Intelligent and Low-carbon Technology Application
{tianleliu, cgn}@post.usts.edu.cn
fan.lyu@cripac.ia.ac.cn    fuyuanhu@mail.usts.edu.cn

## Abstract

Test-Time Discovery (TTD) addresses the critical challenge of identifying and adapting to novel classes during inference while maintaining performance on known classes, which is a capability essential for dynamic real-world environments such as healthcare and autonomous driving. Recent TTD methods adopt training-free, memory-based strategies but rely on frozen models and static representations, resulting in poor generalization. In this paper, we propose a Discrepancy-Amplifying Adapter (DAA), a trainable module that enables real-time adaptation by amplifying feature-level discrepancies between known and unknown classes. During training, DAA is optimized using simulated unknowns and a novel warm-up strategy to enhance its discriminative capacity. To ensure continual adaptation at test time, we introduce a Short-Term Memory Renewal (STMR) mechanism, which maintains a queue-based memory for unknown classes and selectively refreshes prototypes using recent, reliable samples. DAA is further updated through self-supervised learning, promoting knowledge retention for known classes while improving discrimination of emerging categories. Extensive experiments show that our method maintains high adaptability and stability, and significantly improves novel class discovery performance. Our code is available at https://github.com/LeTianL-TT/DAA-for-TTD.

## 1 Introduction

Test-Time Discovery (TTD) [22] is an emerging and increasingly important task that aims to dynamically identify and classify novel categories during the test phase, while simultaneously maintaining robust performance on previously learned classes. This capability is critical for real-world applications such as healthcare, autonomous driving, and robotics, where models must adapt to previously unseen classes after deployment. While Test-Time Adaptation (TTA) [36, 6, 32, 7] has attracted substantial attention for mitigating domain shifts, it typically overlooks the challenge of class shifts, particularly the emergence of new categories. Novel Class Discovery (NCD) [27, 39] is designed for static settings with separated labeled and unlabeled data, and falls short in test-time scenarios, thus generally relies on offline clustering and post-hoc evaluation, assuming that novel categories are only identified in a controlled training environment. *Unlike TTA and NCD, TTD explicitly targets novel class discovery during inference, operates in a distinct paradigm, requiring real-time adaptation*

---

[*]Corresponding authors.

39th Conference on Neural Information Processing Systems (NeurIPS 2025).

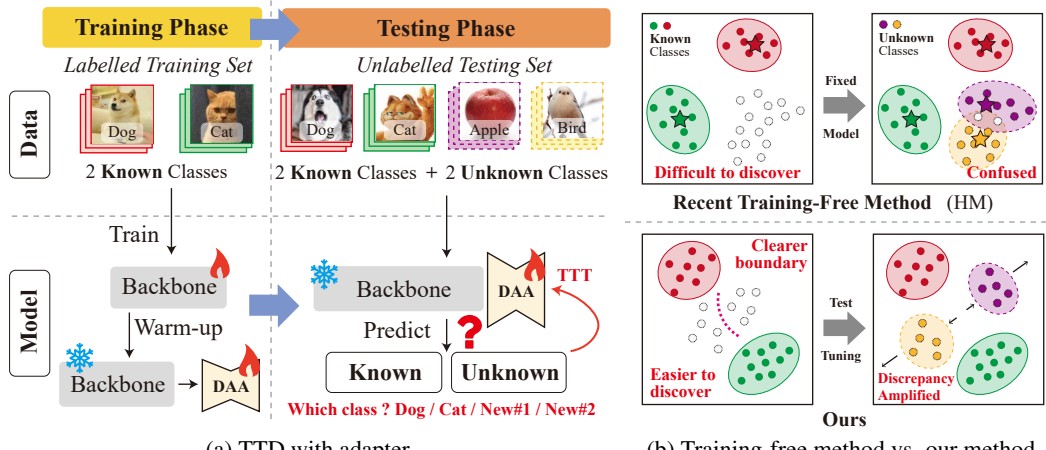

(a) TTD with adapter      (b) Training-free method vs. our method

Figure 1: (a) In Test-Time Discovery (TTD), a model is pretrained on data containing only known classes. During deployment, the test data may include both known and unknown classes, requiring the model to predict known classes and discover novel ones. (b) Recent training-free method [22] relies on fixed features and is difficult to classify via prototype only. Our method maintains known classes and amplifies differences of unknown classes.

*to continuous unlabeled streams of both known and unknown classes, enabling models to maintain performance and learn emerging categories.*

Recent advances in TTD [22] propose a training-free, hash-based memory mechanism for fine-grained comparisons with past samples, enabling novel class discovery during inference. While eliminating the need for model update, this approach relies on frozen model parameters and updates only an external memory. As a result, it suffers from significant estimation errors and limited generalization. As shown in Fig. 1b, the use of static prototypes constrains novel classes to the existing feature space of known classes, often leading to feature overlap and misclassification.

In this paper, we propose a Discrepancy-Amplifying Adapter (DAA) for the TTD task. Unlike prior methods based on frozen architectures, DAA is a trainable module that enables effective adaptation to novel knowledge. During training, the backbone learns known classes, while DAA is optimized to preserve feature consistency. To enhance discrimination of unknown classes, DAA is trained with simulated unknowns, encouraging amplified feature discrepancies by a warm-up strategy. At test time, we introduce a novel Short-Term Memory Renewal (STMR) strategy, inspired by the memory strategy in the previous method [22], to mitigate misclassification. STMR employs a queue-based memory to prioritize recent and reliable samples, supporting prototype updates, sample replay, and test-time refinement of DAA. This design improves both memory efficiency and model adaptability. DAA adapts in real-time testing through self-supervised updates based on predictions and the memory queue, which ensures knowledge alignment for known classes and improves model plasticity and generalization to novel classes. Experimental results show that our method outperforms existing state-of-the-art methods with higher adaptability and stability, and significantly improves novel class discovery performance.

In summary, our contributions are:

(1) Unlike existing training-free methods that rely solely on pre-trained models, we propose a trainable module DAA, to address the TTD problem by amplifying feature differences.

(2) We propose a STMR strategy, which rectifies outdated and unreliable knowledge and is used to maintain known classes and improve the discrimination of unknown classes.

(3) Extensive experiments demonstrate that our method significantly outperforms existing methods.

## 2 Related Work

**Test-Time Training (TTT)** [16, 30, 21] is an emerging paradigm in machine learning that focuses on adapting pre-trained models to distribution shifts during inference. Unlike traditional training methods that rely solely on a fixed dataset, TTT leverages self-supervised tasks to update model parameters

dynamically using the input data structure. TTT methods such as TTT++ [20] and TTT-MAE [9] employ auxiliary tasks like image rotation prediction or masked autoencoding to refine feature extractors during inference. These methods aim to align the model's internal representations with the test data distribution, thereby enhancing generalization [17, 18, 31]. However, TTT methods typically focus on predefined categories and overlook the discovery of novel classes during testing, which is a critical limitation in open-world environments where new classes may emerge unexpectedly.

**Novel Category Discovery (NCD)** [10, 43, 45] addresses the challenge of identifying previously unseen classes during inference. NCD methods can detect new categories autonomously, often through clustering techniques. For instance, DTC [11] leverages prior knowledge of related image classes to reduce ambiguity in clustering and enhance the quality of newly discovered classes. Other methods, such as PromptCCD [3] and GCD [34], employ advanced techniques to improve the detection and representation of novel classes. On-the-fly Category Discovery (OCD) [8, 44] tries to perform online discovery and make instant inference by hash coding and hamming distance. However, OCD focuses on discovery rather than improving performance through the testing phase and can only provide hash descriptions for class prototypes, which ignore the model adaptation on new classes. While NCD has made significant progress in offline settings, it falls short in dynamic, real-time applications where models must adapt to new classes instantly.

**Test-Time Discovery (TTD)** introduces a novel task that enables models to dynamically discover novel classes during inference while maintaining known-class performance, bridging the gap between TTT and NCD. The pioneering work by Lyu et al. [22]. proposes a training-free framework leveraging Locality-Sensitive Hashing (LSH) to construct a memory buffer for efficient sample comparison and pseudo-label refinement. Their method groups similar test samples, combining global prototypes and local hash-based predictions. While this approach mitigates catastrophic forgetting by freezing model parameters, it inherently limits the model's capacity to adaptively refine feature representations for enhanced discrimination between old and new classes. Recent efforts in Test-Time Adaptation (TTA) [1, 4, 25] explore lightweight parameter updates but remain confined to domain shifts rather than class shifts. Our method aims to address the limitations of existing TTD techniques by enhancing the model's ability to adapt to new classes in real-time while maintaining robust performance on known classes, a critical improvement over static architectures in evolving open-world settings.

## 3 Method: Discrepancy-Amplifying Adapter

### 3.1 Problem Definition

The goal of TTD is to enable testing models to not only classify samples from known classes accurately, but also to identify and classify samples from unknown classes dynamically by adaptation. During the *training phase*, let $\mathcal{D}_{\text{train}} = \{(x_i, y_i)\}_{i=1}^{N_{\text{train}}}$ be the training dataset, where $x_i$ is a data point and $y_i \in \mathcal{Y}_{\text{kn}}$ is its corresponding label from the set of **known** classes $\mathcal{Y}_{\text{kn}}$. The model $f$ is trained on $\mathcal{D}_{\text{train}}$ to learn the representations of the known classes. During the *test phase*, the model encounters a test dataset $\mathcal{D}_{\text{test}} = \mathcal{D}_{\text{test}}^{\text{kn}} \cup \mathcal{D}_{\text{test}}^{\text{un}}$. The set of **unknown** classes comprises **seen** and **unseen** parts, i.e., $\mathcal{Y}_{\text{un}} = \mathcal{Y}_{\text{seen}} \cup \mathcal{Y}_{\text{unseen}}$. Once a new class is discovered, it becomes a seen class. As the example shown in Fig. 1a, after discovering new classes "Apple" and "Bird", a TTD model needs to distinguish whether the test sample is "Dog", "Cat", "Apple", or "Bird".

The TTD task requires the model to achieve two objectives. First, *novel class discovery and learning*, where inputs $x \in \mathcal{D}_{\text{test}}^{\text{un}}$ must be identified as belonging to unseen classes and assigned new labels to distinguish them from known-class samples. Second, *unified classification*, where all inputs $x \in \mathcal{D}_{\text{test}}$ are accurately classified into either a known class $y \in \mathcal{Y}_{\text{kn}}$ or a novel class $y \in \mathcal{Y}_{\text{seen}}$, with $\mathcal{Y}_{\text{seen}}$ denoting the set of dynamically discovered novel classes.

Existing TTD methods [22] follow the NCD paradigm, which relies on a fixed backbone where feature representations for unknown classes remain static during the testing phase. These approaches avoid model updates and instead depend on memory buffers storing representative samples, with prediction and novel class discovery based solely on distances to stored prototypes. However, the frozen representations hinder the model's ability to effectively discriminate between known and emerging unknown classes. To overcome this limitation, we introduce a trainable component, the **Discrepancy-Amplifying Adapter** (DAA). Moreover, to improve adaptability and ensure long-term efficacy, we propose **Short-Term Memory Renewal** (STMR), a dynamic memory mechanism that continuously updates and refreshes stored representations.

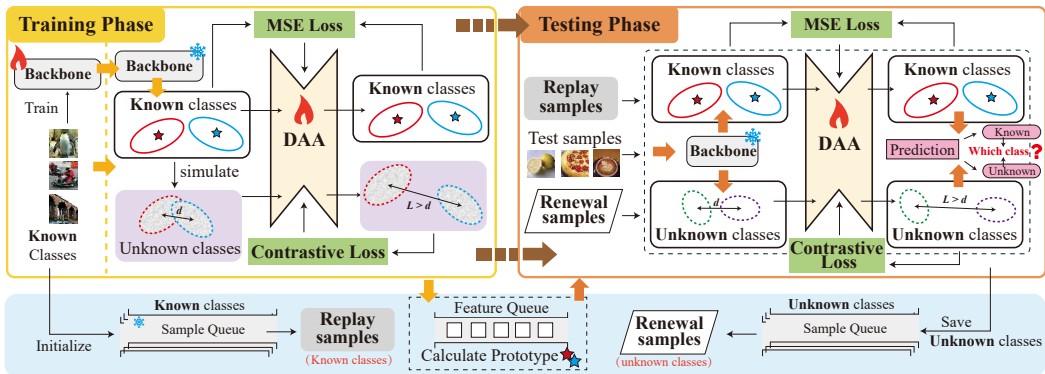

Figure 2: Method Schema. The DAA preserves known class features and amplifies discrepancies with unknowns—simulated during training and encountered during testing. It is updated online to maintain discrimination. The memory system includes sample queues for replay and renewal, and feature queues for prototype computation.

## 3.2 Discrepancy-Amplifying Warm-up before Testing

The failure of past TTD methods was because unknown classes and known classes could not be effectively separated, and this situation could not be improved over time. DAA is a lightweight module inserted into a pre-trained model to enable efficient fine-tuning for unknown classes. Given an input image $x$, the backbone produces a feature vector $\mathbf{r}(x) = f(x) \in \mathbb{R}^D$, which is then transformed by DAA into an adapted representation. The core design of DAA lies in keeping the backbone that contains known classes unchanged, and entrusting the discovery and learning of new classes to the constructed DAA module. To achieve this, we first propose a discrepancy-amplifying warm-up strategy for DAA update before the testing phase. Warm-up has been used in TTA [26, 41] to further build a knowledge structure for the source model [2, 5]. The discrepancy-amplifying warm-up strategy first simulates unknown classes and then trains with a pre-amplify loss.

**Unknown Class Simulation**. During training, only known classes are available, yet the model must remain adaptable to future unknowns. To address this, in the discrepancy-amplifying warm-up strategy, we propose to simulate unknown class features by combining Gaussian noise and the Mixup technique [42]. Given two difference samples $x$ and $x'$, the synthetic feature $\tilde{\mathbf{r}}$ is generated as follows:

$$\tilde{\mathbf{r}}(x) = \lambda \cdot \mathbf{r}(x) + (1 - \lambda) \cdot \mathbf{r}(x') + \mathbf{n}, \tag{1}$$

where $\mathbf{n}$ denotes a vector of Gaussian noise with a mean of zero and a certain covariance matrix. The mixing coefficient $\lambda$ is drawn from a Beta distribution. This augmentation increases training diversity and encourages the DAA to learn more generalized, transferable representations.

**Discrepancy Pre-Amplification**. Building upon the unknown class simulation introduced in the warm-up stage, we design a dual-objective training to guide the discrepancy pre-amplification of DAA. The effect of the training must retain the original representation of known classes while amplifying the distinction between simulated unknown and known features. To preserve the learned semantics of known classes, we constrain the DAA output to remain close to the original backbone feature $f(x)$ using an MSE loss. Simultaneously, given a test batch $\mathcal{B}$, and for $x \in \mathcal{B}$, we apply a contrastive loss on the adapted unknown features to maximize inter-class discrepancy, ensuring the model learns more discriminative representations for novel classes:

$$\mathcal{L}_{\text{kn}}(x) = \|\text{DAA}(\mathbf{r}(x)) - \mathbf{r}(x)\|^2, \ \mathcal{L}_{\text{un}}(x) = -\log \frac{\exp(\sigma[\text{DAA}(\tilde{\mathbf{r}}(x)), \text{DAA}(\tilde{\mathbf{r}}(x))_{aug}])}{\sum_{x' \in \mathcal{B}} \exp(\sigma[\text{DAA}(\mathbf{r}(x')), \text{DAA}(\tilde{\mathbf{r}}(x'))])}, \tag{2}$$

where $\sigma$ denotes the cosine similarity. $\text{DAA}(\tilde{\mathbf{r}}(x))_{aug}$ is the feature obtained through data augmentation operations. The total warm-up loss for DAA is a weighted combination:

$$\mathcal{L}_{\text{train}}(\mathcal{B}) = \mathbb{E}_{x \in \mathcal{B}} \ \mathcal{L}_{\text{kn}}(x) + \mathcal{L}_{\text{un}}(x). \tag{3}$$

This Discrepancy-Amplifying Warm-up strategy has been empirically shown to facilitate novel class discovery by enhancing feature separation during the early training phase. However, during testing, samples from unknown classes often exhibit high uncertainty and noise. Blindly updating the model with such unreliable samples can undermine the benefits of warm-up, potentially causing semantic drift and loss of discrimination between known and unknown classes.

### 3.3 Short-Term Memory Renewal

Inspired by [22], we introduce a Short-Term Memory Renewal (STMR) mechanism during testing. This memory-guided strategy enables the DAA to selectively integrate informative samples while continuously refreshing its representation space, thereby preserving class boundaries and stabilizing adaptation in the presence of noisy unknown inputs. As illustrated in Fig. 3, STMR employs a queue-based memory system consisting of sample queues $\mathcal{S}$ and feature queues $\mathcal{F}$ for each class. For known classes, memory queues are initialized using training data and remain static. Their corresponding prototypes are computed and fixed during the training phase. To prevent catastrophic forgetting, we employ sample replay by randomly selecting $\hat{x}_{kn}^c \sim \mathcal{S}_{kn}^c$ and using them to regularize the DAA updates for known class $c \in \mathcal{Y}_{kn}$. In contrast, novel classes begin with empty memory, which is updated dynamically as new unknown samples are discovered. Unknown-class memory $\mathcal{S}_{un}$ and $\mathcal{F}_{un}$ follows a First-In-First-Out (FIFO) strategy to ensure temporal relevance.

However, since the initial predictions for unknown samples may be noisy and DAA parameters evolve over time, prototype drift and representation mismatch may occur. To address this, STMR introduces a renewal step and it will trigger every several batch. A subset of unknown memory samples $x \sim \mathcal{S}_{un}^c$ are extracted by the backbone and DAA for a seen class $c$. Then, the feature $\hat{\mathbf{r}} = \text{DAA}[\mathbf{r}(x)]$ is re-evaluated to obtain updated predictions $\hat{y}$. If $\hat{y} \in \mathcal{Y}_{kn}$, the sample $x$ is discarded to avoid contamination of unknown memory with misclassified known-class data. If $\hat{y} \in \mathcal{Y}_{un}$, the feature $\hat{\mathbf{r}}$ is used to update DAA via contrastive loss to enhance its separability from known-class features. Then $x$ and $\hat{\mathbf{r}}$ re-queue the sample and its updated feature into $\mathcal{S}_{\hat{y}}$ and $\mathcal{F}_{\hat{y}}$ respectively. The update of the unknown sample and feature queues are updated as follows:

$$\mathcal{S}_{un}^{c=\hat{y}} \leftarrow \text{FIFO}(\mathcal{S}_{un}^{c=\hat{y}}, x), \quad \mathcal{F}_{un}^{c=\hat{y}} \leftarrow \text{FIFO}(\mathcal{F}_{un}^{c=\hat{y}}, \hat{\mathbf{r}}). \tag{4}$$

The prediction and test-time training using DAA can be seen in Sec. 3.4.

For a newly discovered class $c$, the prototype is computed as the mean of its feature queue:

$$\mathbf{p}_c = \mathbb{E}_{x \in \mathcal{S}_{un}^c}(\text{DAA}[\mathbf{r}(x)]). \tag{5}$$

Compared with the memory strategy in HM [22], which stores uncertain samples over time but does not update the model, STMR uses short-term, renewable memory with selective filtering to retain only reliable representations. This design not only avoids the accumulation of errors but also enables model updates via contrastive learning, allowing STMR to adapt to novel classes while preserving known-class performance.

### 3.4 Test-Time Prediction and Training with DAA

In this subsection, we illustrate how to use DAA to conduct prediction and learning at test time. When encountering the test data point with a known or unknown class, the prediction procedure involves calculating the cosine similarity between the test samples and the prototypes of known classes to determine their classification. Based on the max similarity scores, we apply a confidence threshold $\gamma$ to determine whether a test sample $x$ belongs to a seen class or an unseen class.

$$\hat{y} = \begin{cases} \arg\max_{c \in (\mathcal{Y}_{kn} \cup \mathcal{Y}_{seen})}(P(x)), & \text{if } \max_c P(x) > \gamma, \\ \text{new unseen class}, & \text{otherwise}, \end{cases} \tag{6}$$

where prediction comparison $P(x) = \text{sim}(\text{DAA}[\mathbf{r}(x)], \mathbf{p}_c)$, $\text{sim}()$ operator refers to the similarity (cosine similarity) between two vectors. If the sample is classified as an unseen class $\hat{y} \in \mathcal{Y}_{unseen}$, this class will become a seen class $\mathcal{Y}_{seen} \leftarrow \hat{y}$, and the model will treat this sample as the first instance of this new seen class. The feature vector $\text{DAA}[\mathbf{r}(x)]$ will be used as the initial prototype for this class, denoted as $\mathbf{p}_{\hat{y}}$. Subsequent samples will be compared to all known and seen prototypes, including this new $\mathbf{p}_{\hat{y}}$.

For the test-time training, we update the DAA to involve knowledge from novel classes. For known classes, we continue to use the MSE loss, which penalizes deviations between the original and adapted features to ensure that the DAA does not alter the feature representations significantly. Converse to known classes, unknown classes are handled with a contrastive loss that encourages the DAA to produce features distinct from both known classes and other unknown classes. Specifically, given a test batch data $\mathcal{B}$, the test-time training with DAA can be represented by two kinds of loss functions.

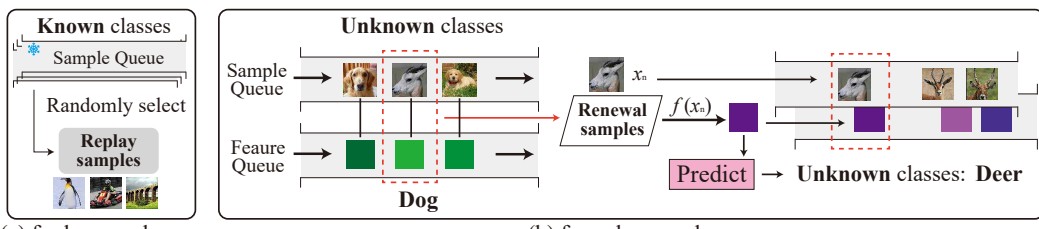

(a) for known classes        (b) for unknown classes

Figure 3: STMR strategy. Replay samples are randomly drawn from the known-class memory to mitigate catastrophic forgetting in DAA, while renewal samples are selected from the unknown-class memory to refresh prototype representations and support timely memory updates.

The first kind of loss is the self-supervised learning loss via the pseudo labels, which is similar to the test-time training in TTA:

$$\mathcal{L}_{\text{kn}}(\mathcal{B}, c) = \mathbb{E}_{x \in \mathcal{B}} \ \mathcal{L}_{\text{kn}}(x | \hat{y} = c), \quad \mathcal{L}_{\text{un}}(\mathcal{B}, c) = \mathbb{E}_{x \in \mathcal{B}} \ \mathcal{L}_{\text{un}}(x | \hat{y} = c). \tag{7}$$

The second kind represents the replay for known and the renewal for seen classes via retrain samples in memory:

$$\mathcal{L}_{\text{kn}}^{\text{replay}}(c) = \mathbb{E}_{(x, \hat{\mathbf{r}}) \in (\mathcal{S}_{\text{kn}}^c, \mathcal{F}_{\text{kn}}^c)} \ \mathcal{L}_{\text{kn}}(x), \quad \mathcal{L}_{\text{un}}^{\text{renewal}}(c) = \mathbb{E}_{(x, \hat{\mathbf{r}}) \in (\mathcal{S}_{\text{un}}^c, \mathcal{F}_{\text{un}}^c)} \ \mathcal{L}_{\text{un}}(x). \tag{8}$$

The overall testing phase loss for the DAA is shown below:

$$\mathcal{L}_{\text{test}}(\mathcal{B}) = \lambda_1 \cdot \mathbb{E}_{c \in \mathcal{Y}_{\text{kn}}} \left( \mathcal{L}_{\text{kn}}(\mathcal{B}, c) + \mathcal{L}_{\text{kn}}^{\text{replay}}(c) \right) + \lambda_2 \cdot \mathbb{E}_{c \in \mathcal{Y}_{\text{seen}}} \left( \mathcal{L}_{\text{un}}(\mathcal{B}, c) + \mathcal{L}_{\text{un}}^{\text{renewal}}(c) \right), \tag{9}$$

where $\lambda_1$ and $\lambda_2$ are hyperparameters that balance the contributions of each loss term.

**Discussion: Training-free TTD vs. DAA**. Training-free TTD approaches such as HM [22] rely only on a fixed model and static feature distribution, updating only class prototypes to accommodate novel classes. While effective in certain scenarios, such methods struggle with complex or overlapping feature distributions. In particular, when unlearned features from different unknown classes are entangled, prototypes are constrained to represent mixed semantics, resulting in suboptimal discrimination and degraded classification performance. Moreover, the absence of model adaptation during test time limits their capacity to incorporate new knowledge. In contrast, our DAA enables dynamic model updates during testing, allowing the feature distribution to evolve with incoming data. This adaptability improves the model's ability to disentangle and separate unknown classes while preserving representations of known ones. By explicitly amplifying inter-class discrepancies and refining the feature space through continual adaptation, DAA achieves more robust generalization and higher accuracy in open-world scenarios.

## 4 Experiment

### 4.1 Experimental Details

**Dataset details**. We conduct our experiments based on three benchmark datasets, namely CIFAR100 (C100)[14], Caltech-UCSD Birds-200-2011 (CUB)[35] and Tiny ImageNet[15]. All these datasets are split into known and unknown classes (7:3). The model is trained on the known training set, and tested on the mixture of known and unknown test sets. We follow three transformed datasets used for discovery in TTD work: CIFAR100D, CUB-200D, and Tiny-ImageNetD. The dataset partitioning follows the scheme outlined in Table 1. More details of the dataset construction can be seen in Appendix.

Table 1: Statistic of the used datasets.

| Dataset | Labeled | CIFAR100D | | | CUB-200D | | | Tiny-ImagenetD | | |
|---|---|---|---|---|---|---|---|---|---|---|
| | | Known | Unknown | No. of samples | Known | Unknown | No. of samples | Known | Unknown | No. of samples |
| TrainSet | ✓ | 70 | 0 | 35000 | 140 | 0 | 4195 | 140 | 0 | 70000 |
| TestSet | | 70 | 30 | 10000 | 140 | 60 | 5794 | 140 | 60 | 10000 |

Table 2: Major comparisons on CIFAR100D, CUB-200D, and Tiny-ImageNetD (*Tiny-IND in table*). Real-time evaluation reflects the accumulated performance across all test batches, while post evaluation reassesses all test samples after training the DAA, updating the memory and prototypes. (**Bold** data is the best performance and Underline data is the second-best performance.)

| Method | Real-time Evaluation | | | | | Post Evaluation | | | | | |
|---|---|---|---|---|---|---|---|---|---|---|---|
| | KA↑ | HCA↑ | ARI↑ | NMI↑ | VM↑ | KA↑ | HCA↑ | ARI↑ | NMI↑ | VM↑ | KF↓ |
| **CIFAR100D** | | | | | | | | | | | |
| Threshold | 76.46±0.98 | 0.64±0.01 | 0.42±0.01 | 0.85±0.00 | 0.85±0.00 | 76.62±1.85 | 0.60±0.01 | 0.42±0.01 | 0.72±0.01 | 0.72±0.01 | 6.45±1.78 |
| L2P [38] | 59.93±2.15 | 0.53±0.00 | 0.37±0.01 | 0.76±0.01 | 0.76±0.01 | 50.53±7.25 | 0.50±0.02 | 0.37±0.01 | 0.69±0.05 | 0.69±0.05 | 27.85±7.21 |
| DP [37] | 66.09±1.01 | 0.58±0.00 | 0.44±0.01 | 0.80±0.00 | 0.80±0.00 | 56.19±2.00 | 0.55±0.01 | 0.44±0.01 | 0.72±0.00 | 0.72±0.00 | 29.06±1.99 |
| GMP [3] | 72.77±1.20 | 0.59±0.00 | 0.31±0.02 | 0.78±0.00 | 0.78±0.00 | 67.21±2.53 | 0.58±0.02 | 0.46±0.01 | 0.71±0.00 | 0.71±0.00 | 17.69±2.53 |
| PHE [44] | 68.18±1.12 | 0.60±0.01 | 0.44±0.01 | 0.73±0.01 | 0.73±0.01 | 68.20±1.07 | 0.58±0.01 | 0.42±0.00 | 0.70±0.01 | 0.70±0.01 | 2.15±0.92 |
| HM [22] | 79.17±**0.13** | 0.61±0.01 | 0.43±0.01 | 0.82±0.01 | 0.82±0.01 | **80.73±1.59** | 0.63±0.00 | 0.48±0.01 | 0.73±0.00 | 0.73±0.00 | 3.41±1.49 |
| Ours | **80.81±0.44** | **0.66±0.01** | **0.52±0.01** | **0.85±0.00** | **0.85±0.00** | 80.27±0.60 | **0.65±0.01** | **0.53±0.01** | **0.75±0.01** | **0.75±0.01** | **3.10±0.70** |
| **CUB200D** | | | | | | | | | | | |
| Threshold | 66.09±1.20 | 0.53±0.02 | 0.20±0.01 | 0.83±0.01 | 0.83±0.01 | 65.52±3.68 | 0.48±0.00 | 0.20±0.00 | 0.70±0.00 | 0.70±0.00 | **1.61±3.55** |
| L2P [38] | 46.22±1.53 | 0.49±0.01 | 0.28±0.01 | 0.80±0.01 | 0.80±0.01 | 31.97±3.35 | 0.41±0.01 | 0.27±0.01 | 0.71±0.01 | 0.71±0.01 | 42.29±3.14 |
| DP [37] | 53.69±1.24 | 0.55±0.03 | 0.29±0.01 | 0.83±0.01 | 0.83±0.01 | 63.37±3.27 | 0.48±0.02 | 0.24±0.01 | 0.48±0.00 | 0.48±0.00 | 5.85±3.11 |
| GMP [3] | 62.97±1.33 | 0.57±0.03 | 0.29±0.00 | 0.84±0.00 | 0.84±0.01 | 58.11±3.00 | 0.48±0.01 | 0.26±0.01 | 0.71±0.00 | 0.71±0.00 | 5.46±2.77 |
| PHE [44] | 44.66±1.03 | 0.49±0.01 | 0.28±0.00 | 0.82±0.01 | 0.82±0.01 | 44.63±0.95 | 0.49±0.01 | 0.24±0.00 | 0.65±0.01 | 0.65±0.01 | 3.96±1.11 |
| HM [22] | 66.20±0.55 | 0.52±0.01 | 0.34±0.01 | 0.83±0.00 | 0.83±0.00 | 64.42±0.65 | 0.50±0.01 | 0.27±0.01 | 0.70±0.00 | 0.70±0.00 | 4.07±0.47 |
| Ours | **68.09±0.33** | **0.63±0.01** | **0.40±0.01** | **0.88±0.00** | **0.88±0.00** | **66.26±0.39** | **0.58±0.01** | **0.32±0.01** | **0.75±0.00** | **0.75±0.00** | 3.60±**0.39** |
| **Tiny-IND** | | | | | | | | | | | |
| Threshold | 57.53±1.80 | 0.57±0.00 | 0.31±0.01 | 0.85±0.01 | 0.85±0.01 | 52.36±3.10 | 0.24±0.01 | 0.15±0.01 | 0.34±0.00 | 0.34±0.00 | 22.90±3.08 |
| L2P [38] | 46.25±1.41 | 0.51±0.01 | 0.33±0.01 | 0.81±0.00 | 0.81±0.00 | 29.50±3.77 | 0.43±0.01 | 0.33±0.01 | 0.69±0.00 | 0.69±0.00 | 47.97±3.77 |
| DP [37] | 46.51±**0.58** | 0.51±0.00 | 0.33±0.00 | 0.81±0.00 | 0.81±0.00 | 28.53±3.33 | 0.42±0.01 | 0.32±0.01 | 0.68±0.01 | 0.68±0.01 | 47.57±3.32 |
| GMP [3] | 62.47±1.40 | 0.51±0.01 | 0.30±0.01 | 0.72±0.01 | 0.72±0.01 | 63.95±2.04 | 0.56±0.01 | 0.43±0.01 | 0.72±0.01 | 0.72±0.01 | 16.86±2.04 |
| PHE [44] | 58.39±1.29 | 0.55±0.02 | 0.25±0.00 | 0.86±0.01 | 0.86±0.01 | 58.39±1.14 | 0.45±0.01 | 0.34±0.01 | 0.64±0.01 | 0.64±0.01 | 3.47±1.32 |
| HM [22] | 75.31±1.31 | 0.61±0.00 | 0.38±0.00 | 0.87±0.00 | 0.87±0.00 | 74.94±2.20 | 0.56±0.02 | 0.40±0.00 | 0.72±0.00 | 0.72±0.00 | **1.15±2.18** |
| Ours | **76.38±**0.82 | **0.63±0.01** | **0.41±0.01** | **0.88±0.00** | **0.88±0.00** | **75.50±1.96** | **0.58±0.01** | **0.42±0.01** | **0.73±0.00** | **0.73±0.00** | 2.39±1.56 |

**Implementation details**. In our implementation, we build our method on the prompt-based method L2P [38], which employs a ViT-B/16 backbone [13] following the pretraining procedure of NCD and GMP work. We employed the contrastive loss of the GCD literature when we fine-tune the pretrained model on the known classes, using SGD optimizer and cosine decay learning rate scheduler with an initial learning rate of 0.1 and minimum learning rate of 0.0001, and weight decay of 0.00005. All input images are resized to $224 \times 224$ and augmented to match the pretrained backbone settings.

**Evaluation metrics**. Following [22], we use the following metrics. We first provide some cluster metrics that traditional NCD methods use, including Hungarian Cluster Accuracy (HCA) [24], Adjusted Rand Index (ARI) [28], Normalized Mutual Information (NMI) [23] and V-Measure [29]. (1) measures the clustering accuracy by optimal one-to-one mapping between predicted clusters and true labels using the Hungarian algorithm. (2) **ARI** quantifies the similarity between the predicted clustering assignments and the true labels while adjusting for chance. (3) **NMI** assesses the mutual dependence between predicted and true labels by shared information between the two distributions. (4) **VM** simultaneously constrains the purity and coverage of the clusters through the harmonic mean. For known classes, we also employ two key metrics to comprehensively assess the model's performance: *Known Accuracy* (**KA**) and *Known Forgetting* (**KF**). KA measures the traditional classification accuracy of the model on known classes while KF quantifies the degree of performance degradation on known classes over time.

For unknown classes, we use two agreement metrics: (1) *True-label Agreement ratio* (**TA**). This metric measures the maximum proportion of samples from a given true class that are predicted as the same class; and (2) *Cluster Agreement ratio* (**CA**). This metric measures the maximum proportion of samples from a given predicted cluster that have the same true label.

$$\text{TA} = \mathbb{E}_{c \in \mathcal{Y}_{\text{seen}}^{\text{GT}}} \frac{1}{|\mathcal{D}_c^{\text{test}}|} \max_{p \in \mathcal{Y}_{\text{seen}}} \left( \sum_{x \in \mathcal{D}_c^{\text{test}}} \mathbf{1}[\hat{y}(x) = p] \right), \tag{10}$$

$$\text{CA} = \mathbb{E}_{p \in \mathcal{Y}_{\text{seen}}} \frac{1}{|\mathcal{C}_p^{\text{test}}|} \max_{c \in \mathcal{Y}_{\text{seen}}^{\text{GT}}} \left( \sum_{(x,y) \in \mathcal{C}_p^{\text{test}}} \mathbf{1}(y = c) \right). \tag{11}$$

where $\mathbf{1}(\cdot)$ is the indicator function (1 if true, 0 otherwise). For more details of the metrics, see Appendix.

## 4.2 Major Comparisons

In this paper, we first compare our methods with naive thresholding-based training-free methods. When exceeding the threshold, the naive method will be considered to have discovered a new class. And we compare with some training-required methods including L2P [38], DP [37], and GMP [3], these methods update prompts like TTA and CL methods. We also compare with the recent PHE [44] method for OCD and HM [22] method for TTD.

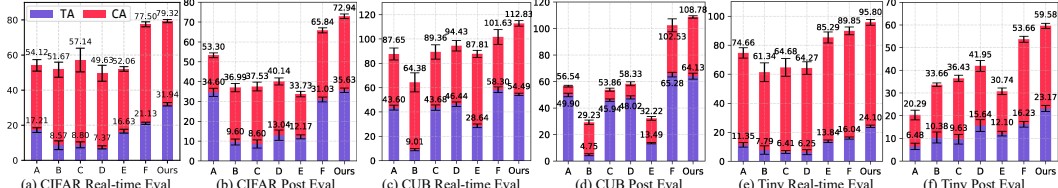

Figure 4: Major comparisons (TA↑, CA↑) on CIFAR100D, CUB-200D, and Tiny-ImageNetD. Real-time evaluation reflects the accumulated performance across all test batches, while post-evaluation reassesses all test samples after the whole test phase. Method A, B, C, D, E and F represent "Threshold", "L2P", "DP", "GMP", "PHE" and "HM".

**Comparisons on clustering metrics**. As shown in Table 2, across multiple datasets, our method demonstrates significant advantages in both real-time and post evaluations. In real-time evaluation, our method achieves the best performance on HCA, ARI, NMI, and VM metrics (e.g., 0.66, 0.52, 0.85, and 0.85 on CIFAR; 0.63, 0.40, 0.88, and 0.88 on CUB; 0.63, 0.41, 0.88, and 0.88 on Tiny-ImageNet). This indicates that our method excels in hierarchical clustering, clustering quality, and matching accuracy. In post evaluation, our method also maintains the highest performance on these metrics (e.g., 0.65, 0.53, 0.75, and 0.75 on CIFAR; 0.58, 0.42, 0.75, and 0.75 on CUB; 0.58, 0.42, 0.73, and 0.73 on Tiny-ImageNet), further validating the superiority of our method in clustering and matching tasks. Additionally, regarding the knowledge forgetting (KF) metric, although knowledge updates may sometimes result in slightly higher KF values compared to methods that do not update the model, our experiments show that the forgetting of existing knowledge during the DAA update process is minimal. There is no forgetting, and our method exhibits good knowledge retention capabilities.

**Comparisons on TA & CA**. In our analysis, TA and CA are employed to quantify the agreement within true labels and clusters, respectively. However, simply predicting most samples into a single new cluster can inflate either TA or CA values. This scenario, while focusing single metric, may not reflect meaningful clustering performance. Therefore, our goal is to achieve a balanced and maximized performance in both TA and CA. This ensures that the clustering results are not only consistent with the true labels but also maintain meaningful and distinct cluster structures. The analysis of our experimental results, as detailed in Fig. 4, reveals several key insights into the performance of various methods in the context of Test-Time Discovery (TTD). The comparison between training-based and training-free methods is particularly illuminating. Training-based methods, which update the model parameters upon encountering new classes, tend to degrade in performance. This is attributed to the immediate adaptation to new classes, which often results in lower TA and CA metrics, along with an increased risk of catastrophic forgetting, where the model loses its ability to recognize previously learned classes. Conversely, training-free methods like HM, which do not update the model parameters, struggle to learn from new classes effectively. This limitation arises because the model's capacity to refine its representations in response to novel classes is constrained. Our proposed method balances these extremes and yields more balanced performance across all three datasets, indicating that our method enhances the network's adaptability in discovering novel classes.

### 4.3 Analysis on DAA

**Distance between prototypes.** In Fig. 5, the prototype distance analysis highlights the superior effectiveness of the DAA method over HM baselines. Specifically, features learned with DAA exhibit larger prototype distances, indicating enhanced class separability, particularly for novel classes. We also compute the ratio of average intra-class to inter-class distances. A higher Intra/Inter ratio reflects tighter within-class clustering and greater between-class separation. As shown, the DAA method yields a significantly higher Intra/Inter ratio than HM methods, demonstrating its strong ability to produce more discriminative and well-structured feature representations for new classes.

**Ablation study**. Table 3 summarizes the ablation results. Activating TTT alone improves CA but leads to model instability and a higher KF. Using only DAA (TA+CA: 76.00 real-time, 65.13 post) outperforms the fixed model (71.84 real-time, 64.34 post), confirming the effectiveness of discrepancy amplification. Combining TTT and DAA further boosts TA and CA in real-time settings, though KF remains high. Incorporating STMR into the full model yields the best overall performance (TA+CA: 78.42 real-time, 73.52 post) and reduces KF, demonstrating that STMR stabilizes prototype updates by mitigating interference from outdated representations and enhancing knowledge retention.

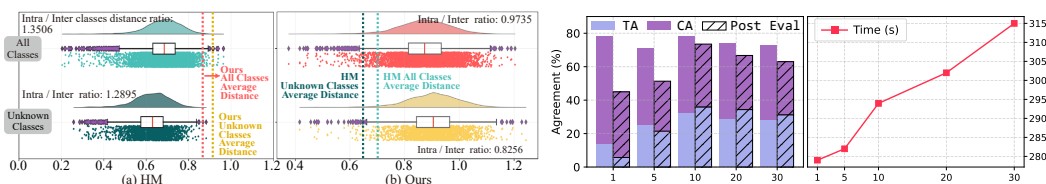

Figure 5: Distance between prototypes.

Figure 6: Different memory size.

**Different Memory Size.** Figure 6 shows the effect of memory size on model performance. While larger memory can store more samples, it may include outdated or irrelevant data, weakening prototype quality. Conversely, too small a memory may fail to capture representative features, reducing generalization. Additionally, increasing memory size leads to higher computational cost. These results highlight the need to balance memory capacity and efficiency to ensure accurate prototype representation without excessive time overhead.

**Different discoverable class numbers.** Given that the distribution of known and unknown classes varies across different scenarios, Table 8 shows that it is crucial to appropriately set the number of discoverable classes to achieve optimal TTD performance. Our method demonstrates superior performance compared to the HM method across various settings. Specifically, when the number of discoverable classes is increased, our method generally achieves higher TA and CA, indicating better adaptation to new classes and more coherent clustering. However, if the number of discoverable classes far exceeds the true number, real-time evaluation shows improvements in TA and CA, but post-evaluation reveals a drop in TA and more severe forgetting indicated by higher KF. While a higher number of discoverable classes can enhance adaptability, it may also introduce noise and disrupt the model's ability to retain knowledge. More experiment is mentioned in Appendix.

Table 3: Ablation study. T: Test-time Training, D: DAA, S: STMR, T+C: TA+CA.

| T | D | S | Real-time Eval | | | Post Eval | | | |
|---|---|---|---|---|---|---|---|---|---|
| | | | TA | CA | T+C | TA | CA | T+C | KF |
| | | | 23.30 | 48.54 | 71.84 | 24.53 | 39.80 | 64.33 | 2.04 |
| ✓ | | | 18.57 | 51.86 | 70.43 | 19.30 | 41.75 | 61.05 | 12.61 |
| | ✓ | | 25.63 | 50.37 | 76.00 | 25.11 | 40.02 | 65.13 | 2.70 |
| ✓ | ✓ | | 30.79 | 42.61 | 73.40 | 36.37 | 33.11 | 69.48 | 4.65 |
| ✓ | ✓ | ✓ | 32.40 | 46.02 | 78.42 | 35.97 | 37.55 | 73.52 | 3.54 |

Table 4: Comparisons of different discoverable class numbers.

| | Known + | Real-time Eval | | | Post Eval | | | |
|---|---|---|---|---|---|---|---|---|
| | Unknown | TA | CA | T+C | TA | CA | T+C | KF |
| **HM** | 70+30 | 21.11 | 56.87 | 77.98 | 31.03 | 34.81 | 65.84 | 3.47 |
| | 80+20 | 11.57 | 64.15 | 75.72 | 20.90 | 31.09 | 51.99 | 0.69 |
| | 90+10 | 14.92 | 52.98 | 67.90 | 21.50 | 30.61 | 52.11 | 0.44 |
| | 70+∞ | 20.37 | 92.94 | 113.31 | 22.63 | 47.09 | 69.72 | 10.69 |
| **Ours** | 70+30 | 32.40 | 46.02 | 78.42 | 35.97 | 37.55 | 73.52 | 3.54 |
| | 80+20 | 36.21 | 48.13 | 84.34 | 32.56 | 31.44 | 64.00 | 1.73 |
| | 90+10 | 41.84 | 30.55 | 72.39 | 31.00 | 27.96 | 58.96 | 1.57 |
| | 70+∞ | 31.78 | 89.58 | 121.36 | 22.35 | 49.32 | 71.67 | 11.07 |

**Frenquency of STMR.** The frequency of STMR plays a crucial role in balancing the trade-off between model performance and computational efficiency. As shown in Fig. 7, A lower frequency of STMR leads to insufficient updates and refinements of the model during the test phase, resulting in poorer recognition performance for both known and novel classes. While increasing the frequency of STMR can improve the model's ability to adapt to new data, it also leads to a substantial increase in computational overhead. Frequent STMR operations require more time and resources for processing each batch.

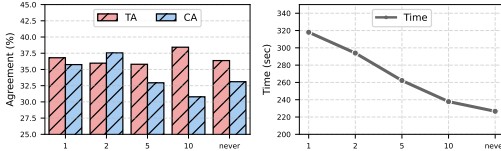

Figure 7: Trends of TA, CA and KF with different STMR frenquency.

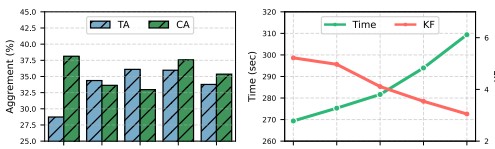

Figure 8: Trends of TA, CA and KF with different replay samples.

**Different replay samples.** The number of replay samples used in the memory replay mechanism affects the performance of our method. Fig. 8 shows that when no replay samples are used, the model exhibits the highest level of knowledge forgetting, as indicated by the largest KF value. As the number of replay samples increases, the KF value decreases at the cost of increased computational time. A well-tuned number of replay samples ensures that the model can effectively leverage memory replay to mitigate catastrophic forgetting while maintaining reasonable processing times.

**Post visualization using t-SNE**. In Fig. 9, we employ t-SNE [33] to visualize the true-label distribution of test samples. L2p and GMP methods that directly update the base model parameters often risk disrupting the model's existing structure, leading to destructive confusion between classes. HM methods, due to their inability to update the model, maintain a static feature distribution. In contrast, our method strikes a balance by actively disrupts the feature distribution and attempts to learn updates that separate features from different classes and keep a clearer boundary than other methods. Effect of DAA in our method can be seen in Appendix.

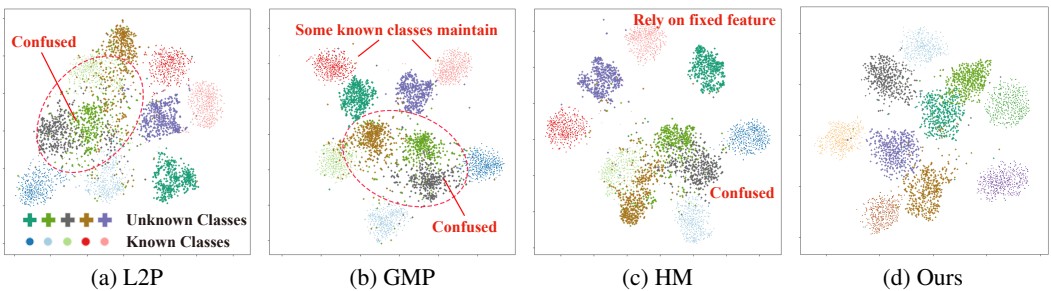

(a) L2P       (b) GMP       (c) HM       (d) Ours

Figure 9: T-SNE visualization on CIFAR100D with 5 known and 5 unknown classes.

**Comparison of unknown class label matching.** Fig. 10 shows the matching between predictions and ground truth, where the base-model updating method easily makes chaos and fails to classify. In the Prototype-only method like HM, lots of old samples are classified into unknown clusters, while DAA w/o STMR performs better. DAA w/ STMR further corrects many classification relationships, increases the number of samples in unknown clusters and reduces the number of samples in old categories that are misclassified.

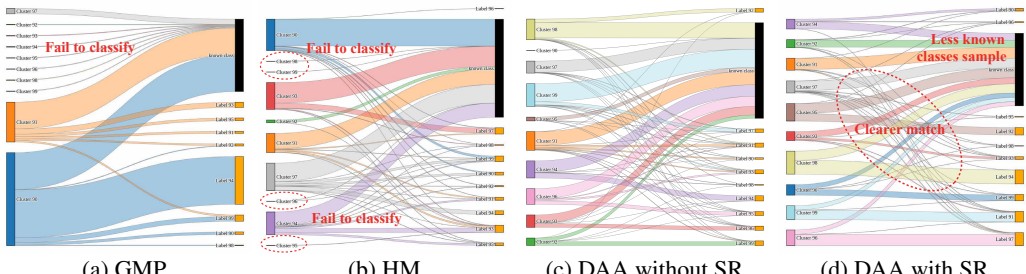

(a) GMP       (b) HM       (c) DAA without SR       (d) DAA with SR

Figure 10: Matching comparison between unknown class predictions and ground truth across methods on CIFAR-100D (90+10). Black rectangles indicate old class samples misclassified as unknowns.

## 5 Conclusion

In this paper, we introduced the Discrepancy-Amplifying Adapter (DAA) and the Short-Term Memory Renewal (STMR) strategy to tackle the challenge of Test-Time Discovery (TTD). By combining trainable adaptation with efficient memory management, our method enables accurate and dynamic identification of novel classes while preserving performance on known classes. DAA enhances feature discrimination by amplifying discrepancies between known and unknown categories, while STMR ensures adaptive and efficient prototype updates through short-term, sample-driven memory refresh. Extensive experiments across multiple benchmarks confirm the superiority of our approach in both real-time and post-hoc evaluations, consistently outperforming existing methods. Our results underscore the importance of real-time adaptability and memory-aware design in TTD. While our method shows strong performance, it introduces additional computation during the warm-up and adaptation phases, and the heuristic-based memory update may limit performance in highly noisy environments. In our future work, we plan to explore more lightweight and adaptive adapter architectures, as well as principled memory selection and compression strategies. Additionally, extending our framework to handle multi-modal or continual learning scenarios could further broaden its applicability in open-world environments.

## Acknowledgements

Our work was supported by the following institutions and projects: National Science and Technology Major Project (No. 2022ZD0117901), National Natural Science Foundation of China (NOs. 62476189, 62406323), China Postdoctoral Science Foundation (No. 2024M753496) and The Postdoctoral Fellowship Program of CPSF (No. GZC20232993). Suzhou's key core technology "list hanging marshal" project (No. SYG2024149), Jiangsu Province Graduate Research and Practical Innovation Plan (No. KYCX25_3572).

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

# DAA: Amplifying Unknown Discrepancy
# for Test-Time Discovery
## ( Appendix )

## A   Comparison of TTD task and other Category Discovery task

We compare TTD with related settings. Out-of-Distribution (OOD) [40, 19] detection neither discovers novel classes nor adapts during inference. TTA [1] handles distribution shifts via self-supervised learning but assumes all test samples belong to known classes. NCD [11] and GCD [34] both aim to identify unknown classes at test time under an offline inference paradigm through clustering the entire test set. OCD [8] resembles TTD in identifying known and unknown classes, but does not incorporate test-time learning from novel classes.

Test-Time Discovery (TTD) is a challenging task that focuses on class shifts rather than domain shifts during test time. It requires the model to not only discover new classes but also classify them accurately while maintaining robust performance on previously seen classes. This dual requirement is particularly demanding due to several intrinsic complexities: the intricate nature of class discovery, the scarcity of labeled data for new classes, and the often ambiguous boundaries between classes. The key challenges in TTD can be summarized as follows:

(1) Distinguishing between the discovery of new classes and the identification of already discovered ones.
(2) Learning and adapting to new classes with limited sample sizes during testing phase.
(3) Avoiding catastrophic forgetting, where the process of learning new classes can inadvertently degrade the model's performance on previously learned classes.

Table 5: Comparison between different category discovery settings.

| Type | Train | Test | Discovery | Test-time Learning |
|------|-------|------|-----------|--------------------|
| OOD | Known classes | Shift Known classes | N/A | N/A |
| TTA | Known classes | Shift Known classes | N/A | Shift Known classes |
| NCD | Known classes | Unknown classes | Post | N/A |
| GCD | Known classes | Known classes + Unknown classes | Post | N/A |
| OCD | Known classes | Known classes + Unknown classes | Real-time | N/A |
| TTD | Known classes | Known classes + Unknown classes | Real-time | Unknown classes |

## B   Details of datasets

We conduct our experiments using three widely recognized benchmark datasets: CIFAR100 (C100) [14], Caltech-UCSD Birds-200-2011 (CUB) [35], and Tiny ImageNet [15]. Each of these datasets is systematically partitioned into known and unknown classes. The model undergoes training on the known training set and is subsequently evaluated on a mixed set containing both known and unknown classes. Since the primary objective of these datasets is to facilitate new class discovery, we follow the HM [22] and use the transformed versions as CIFAR100D, CUB-200D, and Tiny-ImageNetD to reflect this adaptation.

The dataset partitioning follows the scheme outlined in Table 6. Specifically, during the training phase, we divide the training set into known and unknown classes based on their class index order. For instance, in CIFAR100, the first 70 classes are designated as known, while the remaining 30 classes are treated as unknown. The supervised training process is then conducted using only the known classes within the labeled training set. More precisely, CIFAR100D consists of classes 0–69 (70 known classes in total), CUB-200D includes classes 0–139 (140 known classes in total), and Tiny-ImageNetD comprises classes 0–139 (140 known classes in total), all of which are utilized for training.

During the test phase, the model is evaluated on the entire unlabeled test set, which includes samples from all categories, enabling new class discovery and classification. While the category labels remain

structured according to the original known-unknown splits (e.g., 70+30 for CIFAR100D and 140+60 for CUB-200D and Tiny-ImageNetD), these labels are only used for metric evaluation and are not provided to the model during inference. This setup ensures a realistic scenario for open-world learning, where the model must autonomously identify and categorize previously unseen classes.

Table 6: Statistic of the used datasets.

| | | CIFAR100D | | | CUB-200D | | | Tiny-ImagenetD | | |
|---|---|---|---|---|---|---|---|---|---|---|
| Dataset | Labeled | Known | Unknown | No. of samples | Known | Unknown | No. of samples | Known | Unknown | No. of samples |
| TrainSet | ✓ | 70 | 0 | 35000 | 140 | 0 | 4195 | 140 | 0 | 70000 |
| TestSet | | 70 | 30 | 10000 | 140 | 60 | 5794 | 140 | 60 | 10000 |

## C  Metric Definition

The evaluation process is structured into two distinct parts: one focusing on known classes and the other on unknown classes. To ensure a comprehensive assessment, we follow HM [22] and employ both real-time evaluation and post-evaluation strategies. For test-time evaluation, real-time performance is a critical factor. Thus, we compute and report real-time scores for all evaluation metrics as the model processes each test sample. This approach provides immediate insights into the model's performance and enables dynamic tracking of classification accuracy and discovery efficiency. Alongside these real-time scores, we present the final accumulated values, which represent the overall average performance across the entire test set. In addition, recognizing that traditional novel class discovery (NCD) methods typically rely on post-evaluation, we also incorporate this approach for comparative analysis. In post-evaluation, all test samples are revalidated collectively after the entire test phase is complete. This post-hoc evaluation allows for a more refined assessment by leveraging the full distribution of test samples, potentially improving class assignment and clustering accuracy. By providing both real-time and post-evaluation scores, we ensure a thorough and balanced evaluation of the model's effectiveness in handling both known and unknown classes.

### C.1  Metrics for known classes

For the evaluation of known classes, we employ two key metrics to comprehensively assess the model's performance: Known Accuracy (KA) and Known Forgetting (KF).

(1) *Known Accuracy* (**KA**). KA measures the traditional classification accuracy of the model on known classes, reflecting its ability to correctly recognize and classify samples that were part of the training set. This metric serves as a standard benchmark for evaluating the retention of previously learned knowledge:

$$\text{KA} = \mathbb{E}_{c \in \mathcal{Y}_{\text{known}}} \frac{1}{|\mathcal{D}_c^{\text{test}}|} \sum_{x \in \mathcal{D}_c^{\text{test}}} \mathbf{1}(\hat{y}(x) = c), \tag{12}$$

where $\mathcal{Y}_{\text{known}}$ is set of predefined known classes, $\mathcal{D}_c^{\text{test}}$ is test samples with ground-truth class $c$, $\hat{y}(x)$ is the predicted label for sample $x$, $\mathbf{1}(\cdot)$ is the indicator function (1 if prediction matches true class $c$, 0 otherwise)

(2) *Known Forgetting* (**KF**). KF, on the other hand, quantifies the degree of performance degradation on known classes over time. It captures the extent to which the model forgets previously learned information as it encounters new data, particularly when adapting to novel classes. A lower KF score indicates better knowledge retention, while a higher score suggests significant forgetting.

$$\text{KF} = \text{KA}_{\text{pre}} - \text{KA}_{\text{post}}, \tag{13}$$

where $\text{KA}_{\text{post}}$ and $\text{KA}_{\text{pre}}$ are the KA computed on all test data with known classes, before and after Testing Phase.

### C.2  Metrics for unknown classes

For unknown classes, since the predicted label space $\mathcal{Y}_{\text{seen}}^{\text{GT}}$ does not match the cluster label space $\mathcal{Y}_{\text{seen}}$, we propose agreement metrics to assess effectiveness. In the test set $\mathcal{D}^{\text{test}}$, a sample $x$ has a

true label $y \in \mathcal{Y}_{\text{seen}}^{\text{GT}}$ and a predicted cluster label $\hat{y}(x) \in \mathcal{Y}_{\text{seen}}$. We define the subset of $\mathcal{D}^{\text{test}}$ with true label $c$ as $\mathcal{D}_c^{\text{test}}$, and the cluster with predicted label $p$ as $\mathcal{C}_p^{\text{test}}$.

(1) *True-label Agreement ratio* (**TA**). This metric measures the maximum proportion of samples from a given true class that are predicted as the same class:

$$\text{TA} = \mathbb{E}_{c \in \mathcal{Y}_{\text{seen}}^{\text{GT}}} \frac{1}{|\mathcal{D}_c^{\text{test}}|} \max_{p \in \mathcal{Y}_{\text{seen}}} \left( \sum_{x \in \mathcal{D}_c^{\text{test}}} \mathbf{1}[\hat{y}(x) = p] \right), \tag{14}$$

where $\mathbf{1}(\cdot)$ is the indicator function (1 if true, 0 otherwise).

(2) *True-label Entropy* (**TE**). This metric measures the average entropy $H(\cdot)$ of the predicted labels for samples with that true class:

$$\text{TE} = \mathbb{E}_{c \in \mathcal{Y}_{\text{seen}}^{\text{GT}}} H(\{\hat{y}(x) | x \in \mathcal{D}_c^{\text{test}}\}) \tag{15}$$

where $H(\cdot)$ is Shannon entropy of predicted label distribution which is used to quantifies uncertainty or diversity in a distribution, $H(z_i) = -\sum_{z \in \mathcal{Z}} q(z) \log_2 q(z)$.

(3) *Cluster Agreement ratio* (**CA**). This metric measures the maximum proportion of samples from a given predicted cluster that are with the same true label:

$$\text{CA} = \mathbb{E}_{p \in \mathcal{Y}_{\text{seen}}} \frac{1}{|\mathcal{C}_p^{\text{test}}|} \max_{c \in \mathcal{Y}_{\text{seen}}^{\text{GT}}} \left( \sum_{(x,y) \in \mathcal{C}_p^{\text{test}}} \mathbf{1}(y = c) \right). \tag{16}$$

(4) *Cluster Entropy* (**CE**). This metric measures the average entropy of the samples that predicted the true class contained in clusters:

$$\text{CE} = \mathbb{E}_{p \in \mathcal{Y}_{\text{seen}}} H(\{y | (x,y) \in \mathcal{C}_p^{\text{test}}\}). \tag{17}$$

## C.3 Clustering metrics

Traditional novel class discovery (NCD) methods typically rely on post-cluster evaluation, where the quality of the discovered clusters is assessed after the entire test set has been processed. To ensure a comprehensive comparison with existing approaches, we also report several widely used clustering evaluation metrics, including Hungarian Cluster Accuracy (HCA) [24], Adjusted Rand Index (ARI) [28], Normalized Mutual Information (NMI) [23], and V-Measure [29]. Note that these metrics are only evaluated after TTD, say post evaluation.

(1) *Hungarian Cluster Accuracy* (**HCA**). This metric measures the clustering accuracy by computing an optimal one-to-one mapping between predicted clusters and ground-truth labels using the Hungarian algorithm. It provides an intuitive evaluation of how well the discovered clusters align with the actual class distributions. HCA can be computed as

$$\text{HCA} = \mathbb{E}_{(x,y) \in \mathcal{D}^{\text{test}}} (y = \text{map}(\hat{y}(x))), \tag{18}$$

where $\text{map}(\cdot)$ is the optimal mapping from clustering to true labels obtained based on the Hungarian algorithm

(2) *Adjusted Rand Index* (**ARI**). ARI quantifies the similarity between the predicted clustering assignments and the ground-truth labels while adjusting for chance. It accounts for both correct pairwise clustering and misclustered pairs, offering a robust measure of clustering consistency.

$$\text{ARI} = \frac{\text{RI} - \mathbb{E}[\text{RI}]}{\max(\text{RI}) - \mathbb{E}[\text{RI}]}, \tag{19}$$

where Rand Index $(\text{RI}) = \frac{a+b}{C_n^2}$, $a$ is the logarithm of samples of the same class assigned to the same cluster, and $b$ is the logarithm of samples of different classes assigned to different clusters. $n$ is the total number of samples, combination $C_n^2 = \frac{n(n-1)}{2}$ and $\mathbb{E}[\text{RI}]$ is the expected value of RI.

(3) *Normalized Mutual Information* (**NMI**). NMI assesses the mutual dependence between predicted and true labels by measuring the shared information between the two distributions. A higher NMI value indicates better alignment between the discovered clusters and the actual categories. The value

interval of NMI is [0,1], and a larger value indicates a higher degree of information sharing between the clustering results and the real labels.

$$\text{NMI}(\mathcal{U}, \mathcal{V}) = \frac{2 \cdot I(\mathcal{U}; \mathcal{V})}{H(\mathcal{U}) + H(\mathcal{V})}, \tag{20}$$

where $\mathcal{U}$ is collection of true labels and $\mathcal{V}$ is collection of predictions. $I(\mathcal{U}; \mathcal{V})$ is Mutual Information where $I(\mathcal{U}; \mathcal{V}) = H(\mathcal{U}) - H(\mathcal{U}|\mathcal{V})$. $H(\mathcal{U})$ is the entropy of true label, $H(\mathcal{U}) = -\sum_{c=1}^{C} p(c) \log p(c)$, and $H(\mathcal{V})$ is the entropy of prediction, $H(\mathcal{V}) = -\sum_{k=1}^{K} p(k) \log p(k)$.

(4) *V-Measure* (**VM**). The VM is taken in the interval [0,1], which simultaneously constrains the purity and coverage of the clusters through the harmonic mean. Both VM and NMI are symmetric metrics that support the comparison of clusters and categories at different scales.

$$\text{V-Measure} = \frac{2 \cdot h \cdot c}{h + c}, \tag{21}$$

where homogeneity $h = 1 - \frac{H(\mathcal{U}|\mathcal{V})}{H(\mathcal{U})}$, and completeness $c = 1 - \frac{H(\mathcal{V}|\mathcal{U})}{H(\mathcal{V})}$. $H(\mathcal{U}|\mathcal{V}) = -\sum_{k=1}^{K} \sum_{t=1}^{T} p(k,t) \log \frac{p(k,t)}{p(k)}$ and $H(\mathcal{V}|\mathcal{U}) = -\sum_{t=1}^{T} \sum_{k=1}^{K} p(t,k) \log \frac{p(t,k)}{p(t)}$, where $p(t) = \frac{N_t}{N}$ is the sample proportion of class $t$, $p(k) = \frac{N_k}{N}$ is the sample proportion of cluster $k$, and $p(t,k) = \frac{count_k(t)}{N}$ is the joint distribution probability.

# D   More TTD Comparisons

In this section, we provide a detailed comprehensive comparison of our method with several approaches on three benchmark datasets: CIFAR100D, CUB-200D, and Tiny-ImageNetD. The evaluation includes both real-time and post evaluations, where real-time evaluation reflects the accumulated performance across all test batches, and post evaluation reassesses all test samples after training the DAA, updating the memory, and prototypes. The results are summarized in Table 7.

## D.1   Comparisons on TA & CA

TA and CA are critical metrics for evaluating the performance of test-time discovery methods. TA measures the overall accuracy of the model in predicting the correct class labels, while CA evaluates the model's ability to correctly classify samples within each class. A balanced performance in both TA and CA indicates that the model not only achieves high overall accuracy but also maintains meaningful and distinct class structures. Analysis is given in the main text.

## D.2   Comparisons on TE & CE

TE and CE are complementary metrics to TA and CA, respectively. Lower values of TE and CE indicate better performance. However, Our method is not training-free, which means it updates the model's representations during test time. This adaptive updating mechanism allows our method to refine its predictions and improve performance. As a result, our method inevitably has higher TE and CE values compared to training-free methods like HM.

And the results demonstrate that our method achieves a balanced and maximized performance in both TA and CA, while also maintaining relatively low TE and CE values. This is attributed to our DAA and STMR mechanism, which allows the model to refine its representations in response to new classes without suffering from catastrophic forgetting. Unlike training-free methods that struggle to learn from new classes, our method leverages the benefits of continuous adaptation to improve performance. Compared to other model-training methods, our method shows superior robustness and adaptability, making it a more effective solution for test-time discovery tasks.

# E   Different discoverable class numbers

In our experiments, we set an upper limit on the number of discoverable classes to investigate its impact on the performance of test-time discovery (TTD). This is a crucial parameter, as real-world scenarios may involve a much larger or even infinite number of potential new classes. The results are

Table 7: More TTD comparisons on CIFAR100D, CUB-200D, and Tiny-ImageNetD (*Tiny-IND in table*). Real-time evaluation reflects the accumulated performance across all test batches, while post evaluation reassesses all test samples after training the DAA, updating the memory and prototypes. (**Bold** data is the best performance and Underline data is the second-best performance.)

| | Method | Real-time Evaluation | | | | | Post Evaluation | | | | | |
|---|---|---|---|---|---|---|---|---|---|---|---|---|
| | | KA↑ | TA↑ | TE↓ | CA↑ | CE↓ | KA↑ | TA↑ | TE↓ | CA↑ | CE↓ | KF↓ |
| CIFAR100D | Threshold | 76.46±0.98 | 17.21±1.33 | 0.52±0.04 | 36.91±3.26 | 2.07±0.41 | 76.62±1.85 | 34.60±2.02 | 1.10±0.04 | 18.70±1.24 | 1.42±0.05 | 6.45±1.78 |
| | L2P [38] | 59.93±2.15 | 8.57±2.49 | 0.60±0.06 | 43.10±4.30 | 1.85±0.18 | 50.53±7.25 | 9.60±1.50 | 0.77±0.12 | 27.39±2.11 | 1.37±0.24 | 27.85±7.21 |
| | DP [37] | 66.09±1.01 | 8.80±1.69 | 0.53±0.08 | 48.34±6.78 | 1.63±0.30 | 56.19±2.00 | 8.68±2.06 | 0.70±0.08 | 28.93±2.25 | **1.34±0.05** | 29.06±1.99 |
| | GMP [3] | 72.77±1.20 | 7.37±0.88 | **0.51±0.05** | 42.26±4.54 | 1.80±0.24 | 67.21±2.53 | 13.04±2.59 | 1.18±0.06 | 27.10±1.77 | 1.55±0.08 | 17.69±2.53 |
| | PHE [44] | 68.18±1.12 | 16.63±1.08 | 0.81±0.03 | 35.43±1.31 | 1.96±0.10 | 68.20±1.07 | 12.17±**1.01** | **1.06±0.03** | 21.56±1.35 | 1.68±0.06 | **2.15±0.92** |
| | HM [22] | 79.17±0.13 | 21.13±0.62 | 0.67±**0.02** | **56.37±1.42** | **1.23±0.08** | **80.73±1.59** | 31.03±1.24 | 1.07±0.02 | 34.81±1.22 | 1.50±0.02 | 3.41±1.49 |
| | Ours | **80.81±0.44** | **31.94±0.98** | 0.76±**0.02** | 45.38±**0.88** | 1.53±0.06 | 80.27±0.60 | **35.63±**1.11 | 1.45±0.06 | **37.31±1.20** | 1.55±0.09 | 4.10±**0.60** |
| CUB200D | Threshold | 66.09±1.20 | 43.60±2.08 | 0.40±0.06 | 44.05±4.96 | 1.46±0.40 | 65.52±3.68 | 49.90±1.33 | 0.81±0.00 | 6.64±0.67 | 0.70±0.00 | **1.61±3.55** |
| | L2P [38] | 46.22±1.53 | 9.01±**0.87** | 0.44±0.02 | 55.37±7.79 | 0.97±0.25 | 31.97±3.35 | 4.75±0.73 | **0.51±0.03** | 24.48±1.65 | **0.62±0.06** | 42.29±3.14 |
| | DP [37] | 53.69±1.24 | 43.68±2.20 | 0.40±0.06 | 45.68±5.88 | 1.50±0.36 | 63.37±3.27 | 45.94±0.91 | 1.69±0.02 | 7.92±1.10 | 1.10±0.03 | 5.85±3.11 |
| | GMP [3] | 62.97±1.33 | 46.44±1.87 | 0.59±0.03 | 47.99±4.34 | 1.49±0.13 | 58.11±3.00 | 48.02±1.20 | 1.53±0.01 | 10.31±1.45 | 0.90±**0.00** | 5.46±2.77 |
| | PHE [44] | 44.66±1.03 | 28.64±1.43 | 0.63±0.02 | **59.17±**2.88 | **0.93±0.03** | 44.63±0.95 | 13.49±**0.43** | 1.28±0.04 | 18.73±1.39 | 1.56±0.12 | 3.96±1.11 |
| | HM [22] | 66.20±0.55 | **58.30±2.37** | **0.35±**0.02 | 43.33±6.10 | 1.92±0.83 | 64.42±0.65 | **65.28±1.78** | 1.02±0.03 | 37.25±4.90 | 1.24±0.22 | 4.07±0.47 |
| | Ours | **68.09±0.33** | 54.49±1.03 | 0.39±**0.01** | 58.34±1.35 | 1.08±0.05 | **66.26±0.39** | 64.13±2.23 | 1.12±0.08 | **44.65±0.88** | 1.29±0.13 | 3.60±0.39 |
| Tiny-IND | Threshold | 57.53±1.80 | 11.35±1.56 | 0.48±0.03 | 63.31±3.55 | 0.66±0.12 | 52.36±3.10 | 6.48±1.40 | **0.37±0.02** | 13.81±2.11 | **0.44±0.02** | 22.90±3.08 |
| | L2P [38] | 46.25±1.41 | 7.79±2.92 | 0.51±0.03 | 53.55±6.47 | 1.33±0.23 | 29.50±3.77 | 10.38±2.42 | 0.89±0.06 | 23.28±**0.79** | 1.37±0.06 | 47.97±3.77 |
| | DP [37] | 46.51±**0.58** | 6.41±0.93 | 0.51±0.03 | 58.27±6.10 | 1.15±0.21 | 28.53±3.33 | 9.63±2.10 | 0.85±0.02 | 26.80±1.38 | 1.33±0.02 | 47.57±3.32 |
| | GMP [3] | 62.47±1.40 | 6.25±1.72 | **0.45±**0.02 | 58.02±4.29 | 1.08±0.14 | 63.95±2.04 | 15.64±2.63 | 1.30±0.03 | 26.31±2.33 | 1.54±0.03 | 16.86±2.04 |
| | PHE [44] | 58.39±1.29 | 13.84±0.90 | 0.48±0.02 | 71.45±3.80 | **0.40±0.03** | 58.39±**1.14** | 12.10±**1.05** | 0.70±**0.00** | 18.64±1.42 | 1.24±**0.01** | 3.47±**1.32** |
| | HM [22] | 75.31±1.31 | 16.04±0.76 | 0.51±**0.00** | **73.81±2.67** | 0.61±0.04 | 74.94±2.20 | 16.23±1.24 | 0.81±**0.00** | **37.43±1.30** | **1.21±0.02** | **1.15±2.18** |
| | Ours | **76.38±0.82** | **24.10±0.80** | 0.54±**0.00** | 71.70±2.02 | 0.73±0.05 | **75.50±1.96** | **23.17±1.35** | 1.14±0.01 | 36.41±1.12 | 1.44±**0.01** | 2.39±1.56 |

summarized in Table 8, where we compare the performance of our method with the HM method under different settings. We find that increasing the number of discoverable classes generally improves TA and CA, but the effect depends on the number of known classes. For example, when the number of known classes is fixed at 70, increasing the number of discoverable unknown classes from 30 to 100 and then to 200, both TA and CA improve significantly. Specifically, for our method, TA increases from 32.40 to 36.21 and then to 41.84, while CA increases from 46.02 to 48.13 and then to 30.55 (note that CA drops slightly when the number of unknown classes becomes very large). Increasing the number of discoverable classes generally leads to higher KF values, indicating more severe forgetting. For example, when the number of discoverable classes increases from 30 to 200, the KF value for our method increases from 3.54 to 8.27.

Compared to the HM method, our method shows more balanced performance across different settings. For instance, when the number of discoverable classes is set to 100, our method achieves a TA of 36.21 and a CA of 48.13, which are significantly more balanced than those of HM (TA: 17.58, CA: 81.28). This indicates that our method is more effective in balancing the discovery of new classes and the recognition of known classes. However, when the number of discoverable classes far exceeds the true number, both methods suffer from performance degradation. For example, in the case of 70 known classes and an infinite number of discoverable unknown classes, our method achieves a TA of 31.78 and a CA of 89.58, while HM achieves a TA of 20.37 and a CA of 92.94. This suggests that while our method is more robust in general, both methods struggle when the number of discoverable classes becomes excessively large.

The results highlight the importance of appropriately setting the number of discoverable classes for optimal TTD performance. While more discoverable classes generally improve the model's ability to recognize new patterns, they also introduce more complexity and risk of forgetting. Our method shows a more balanced performance across different settings, achieving higher TA and CA values while maintaining reasonable TE, CE, and KF values. This demonstrates the effectiveness of our approach in balancing adaptability and stability during test-time discovery.

# F  Architecture of DAA

We insert the lightweight DAA after the frozen ViT backbone to project the 768-d feature into an updated latent space. The architecture of DAA is a standard 2-layer adapter [12] which includes a Linear down-projection layer $768 \to 128$, ReLU Activation and a Linear up-projection layer $128 \to 768$. Our technical contribution is the training strategy that endows this simple structure to

Table 8: Comparisons of different discoverable class numbers.

| Known + | Real-time Eval | | | | Post Eval | | | | |
|---|---|---|---|---|---|---|---|---|---|
| Unknown | TA | TE | CA | CE | TA | TE | CA | CE | KF |
| **HM** 70+30 | 21.11 | 0.66 | 56.87 | 1.27 | 31.03 | 1.07 | 34.81 | 1.50 | 3.47 |
| 70+100 | 17.58 | 0.70 | 81.28 | 0.44 | 25.03 | 1.82 | 40.74 | 1.05 | 6.46 |
| 70+200 | 19.86 | 0.76 | 85.60 | 0.33 | 26.87 | 2.17 | 42.63 | 0.84 | 7.79 |
| 70+$\infty$ | 20.37 | 0.84 | 92.94 | 0.16 | 22.63 | 2.86 | 47.09 | 0.46 | 10.69 |
| 70+Human | 52.10 | 0.48 | 42.96 | 1.68 | 48.27 | 1.19 | 49.44 | 1.24 | 5.81 |
| **Ours** 70+30 | 32.40 | 0.74 | 46.02 | 1.52 | 35.97 | 1.45 | 37.55 | 1.53 | 3.54 |
| 70+100 | 36.21 | 0.87 | 48.13 | 0.69 | 32.56 | 2.05 | 31.44 | 1.22 | 6.73 |
| 70+200 | 41.84 | 0.90 | 30.55 | 0.67 | 31.00 | 2.22 | 27.96 | 0.93 | 8.27 |
| 70+$\infty$ | 31.78 | 0.98 | 89.58 | 0.35 | 22.35 | 2.99 | 49.32 | 0.55 | 11.07 |
| 70+Human | 75.22 | 0.35 | 30.70 | 1.97 | 49.30 | 1.35 | 50.61 | 1.36 | 2.68 |

maintain known class features and amplify discrepancy among unknown class features to gain better open-world test-time discovery behavior.

- Unlike existing training-free TTD methods like HM, our approach can make DAA have better ability to distinguish between known classes and unknown classes during the warm up phase and trainable during the testing phase.
- Compared to traditional methods of updating the entire backbone, we only update DAA and will not damage the entire backbone.

## G  T-SNE visualization of the effect of DAA.

To provide a more intuitive understanding of how our method affects the feature space, we conducted T-SNE [33] visualizations of the feature embeddings before and after applying our DAA (Dynamic Adaptation and Augmentation) mechanism. The results are shown in Fig. 11.

In the pre-training phase, our method attempts to disrupt the feature representations of unknown classes to some extent. This is evident from the visualization in Fig. 11b, where the feature embeddings of unknown classes are more scattered and less well-separated from known classes. This disruption is intentional, as it helps the model to avoid overfitting to the initial feature space and encourages it to adapt more flexibly during the test phase. After test-time training with DAA, the boundaries between known and unknown classes become clearer again, as shown in Fig. 11c. This indicates that our method effectively refines the feature space during test-time training, allowing the model to better distinguish between known and novel classes. In contrast, baseline methods that fix the model and do not change throughout the entire test phase (as shown in Fig. 11a) struggle to adapt to new classes, resulting in less clear boundaries and poorer performance.

This visualization demonstrates the effectiveness of our DAA mechanism in dynamically adapting the feature space during test-time training, leading to improved performance in recognizing both known and novel classes.

## H  Hyper-parameter analysis

### H.1  Frenquency of STMR.

The frequency of STMR plays a crucial role in balancing the trade-off between model performance and computational efficiency. As shown in Fig. 12, we conducted experiments to investigate the impact of varying the frequency of STMR on the overall performance of our method. The results indicate that as the frequency of STMR decreases, both the TA and CA deteriorate significantly. This suggests that a lower frequency of STMR leads to insufficient updates and refinements of the model during the test phase, resulting in poorer recognition performance for both known and novel classes.

On the other hand, while increasing the frequency of STMR can improve the model's ability to adapt to new data, it also leads to a substantial increase in computational overhead. Frequent STMR operations require more time and resources for processing each batch, which can be impractical for

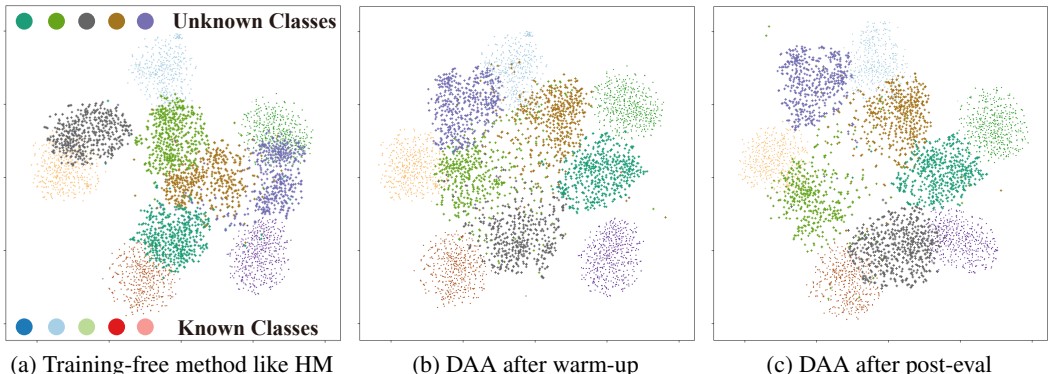

| (a) Training-free method like HM | (b) DAA after warm-up | (c) DAA after post-eval |

Figure 11: T-SNE visualization of the DAA's effect on CIFAR100D with 5 known and 5 unknown classes.

real-time or resource-constrained applications. Moreover, the improvement in performance brought by each additional STMR operation diminishes as the frequency increases, indicating that there is a point of diminishing returns.

Therefore, selecting an optimal frequency for STMR is essential to achieve a balance between performance and efficiency. A moderate frequency ensures that the model can effectively leverage STMR for continuous adaptation while avoiding excessive computational costs.

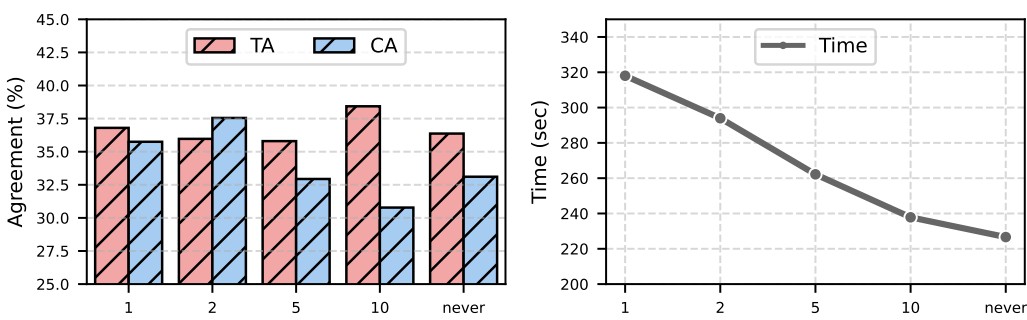

Figure 12: Trends of TA, CA and KF with different STMR frenquency.

## H.2   Different replay samples.

The number of replay samples used in the memory replay mechanism is another critical hyper-parameter that affects the performance of our method. Fig. 13 shows that the number of replay samples has a significant influence on the model's ability to retain knowledge from previous classes while adapting to new ones. When no replay samples are used, the model exhibits the highest level of knowledge forgetting, as indicated by the largest KF value. This suggests that without replay, the model is more prone to catastrophic forgetting, where it quickly forgets previously learned information as it adapts to new data.

As the number of replay samples increases, the KF value decreases, indicating that the model is better able to retain knowledge from previous classes. However, this improvement comes at the cost of increased computational time, as more replay samples require additional processing during each batch. Moreover, while a moderate number of replay samples can help stabilize the model's performance, an excessive number of replay samples can lead to diminishing returns in terms of performance gains, while further increasing the computational burden.

Thus, choosing an appropriate number of replay samples is crucial for achieving a balance between knowledge retention and computational efficiency. A well-tuned number of replay samples ensures that the model can effectively leverage memory replay to mitigate catastrophic forgetting while maintaining reasonable processing times.

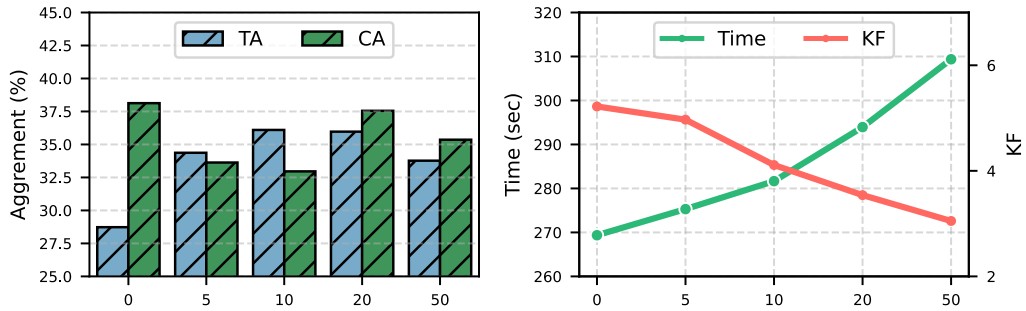

Figure 13: Trends of TA, CA and KF with different replay samples.

## H.3 $\lambda_1$ and $\lambda_2$ during TTT

During test-time training (TTT), the overall testing phase loss for the DAA is shown below:

$$\mathcal{L}_{\text{test}}(\mathcal{B}) = \lambda_1 \cdot \mathbb{E}_{c \in \mathcal{Y}_{\text{kn}}} \left( \mathcal{L}_{\text{kn}}(\mathcal{B}, c) + \mathcal{L}_{\text{kn}}^{\text{replay}}(c) \right) + \lambda_2 \cdot \mathbb{E}_{c \in \mathcal{Y}_{\text{seen}}} \left( \mathcal{L}_{\text{un}}(\mathcal{B}, c) + \mathcal{L}_{\text{un}}^{\text{renewal}}(c) \right), \quad (22)$$

where $\lambda_1$ and $\lambda_2$ are hyperparameters that balance the contributions of each loss term.

We explored the impact of different ratios between $\lambda_1$ and $\lambda_2$ on the overall performance of our method. The results are presented in Table 9. The MSE loss is used to refine the model's predictions for known classes, while the contrastive loss encourages the model to separate the feature embeddings of known and unknown classes. We conducted experiments with different ratios of MSE loss to contrastive loss to determine the optimal balance between these two objectives.

When the model places a much higher emphasis on refining the predictions for known classes. This results in a relatively high TA and CA. However, the model's ability to distinguish between known and unknown classes is somewhat limited, as indicated by the higher CE value. As the ratio decreases, the model starts to pay more attention to the contrastive loss, which helps improve the separation between known and unknown classes. This leads to a slight increase in CA and a decrease in CE, indicating better class separation. However, the overall TA and TE values are affected, suggesting that the model's ability to accurately classify known classes is slightly compromised. When the ratio further decreases, the model crashes, as indicated by the extremely high TA and TE values and the near-zero CA and CE values. This suggests that an excessive emphasis on the contrastive loss can destabilize the model, leading to poor performance.

These results highlight the importance of carefully balancing the MSE loss and contrastive loss during TTT. An optimal ratio ensures that the model can effectively refine its predictions for known classes while also maintaining clear boundaries between known and unknown classes. This balance is crucial for achieving high accuracy and robustness in recognizing both known and novel classes.

Table 9: Comparisons of ratio between $\lambda_1$ and $\lambda_2$ during TTT.

| $\lambda_1 : \lambda_2$ | Real-time Eval | | | | Post Eval | | | | |
| --- | --- | --- | --- | --- | --- | --- | --- | --- | --- |
| | TA | TE | CA | CE | TA | TE | CA | CE | KF |
| 1000:0.1 | 28.41 | 0.80 | 44.91 | 1.56 | 29.50 | 1.49 | 30.53 | 1.74 | 0.92 |
| 1000:0.5 | 29.40 | 0.82 | 45.80 | 1.55 | 33.33 | 1.53 | 34.66 | 1.73 | 1.23 |
| 1000:1 | 33.66 | 0.66 | 49.81 | 1.39 | 34.8 | 1.03 | 37.23 | 1.48 | 2.64 |
| 1000:2 | 32.40 | 0.74 | 46.02 | 1.52 | 35.97 | 1.45 | 37.55 | 1.53 | 3.54 |
| 1000:5(crashed) | 78.22 | 0.20 | 19.00 | 0.38 | 100 | 0.00 | 1.00 | 1.99 | — |
| 500 : 1 | 30.20 | 0.84 | 42.64 | 1.64 | 33.30 | 1.48 | 34.37 | 1.71 | 3.76 |
| 500 : 2 | 34.98 | 0.64 | 49.68 | 1.40 | 37.60 | 1.22 | 32.33 | 1.53 | 4.48 |
| 500 : 3(crashed) | 81.59 | 0.35 | 15.37 | 0.62 | 100 | 0.00 | 1.20 | 2.20 | — |

## H.4 Comparison of different threshold Gamma

During testing phase, we base on the max similarity scores, and apply a confidence threshold $\gamma$ to determine whether a test sample $x$ belongs to a seen class or an unseen class.

$$\hat{y} = \begin{cases} \arg\max_{c \in (\mathcal{Y}_{kn} \cup \mathcal{Y}_{seen})}(P(x)), & \text{if } \max_c P(x) > \gamma, \\ \text{new unseen class,} & \text{otherwise,} \end{cases} \tag{23}$$

where prediction comparison $P(x) = \text{sim}(\text{DAA}[\mathbf{r}(x)], \mathbf{p}_c)$.

$\gamma$ was selected through grid search in our experiment, and we analyzed the impact of $\gamma$. The typical threshold of $\gamma$ is 0.7. And our method is not sensitive to selection from one dataset to other datasets.

Table 10: Comparisons of different $\gamma$ during Testing phase.

| $\gamma$ | Real-time Eval | | | | Post Eval | | | | |
|---|---|---|---|---|---|---|---|---|---|
| | TA | TE | CA | CE | TA | TE | CA | CE | KF |
| 0.5 | 11.85 | 0.83 | 44.36 | 1.61 | 30.03 | 1.39 | 37.14 | 1.67 | 5.89 |
| 0.6 | 20.04 | 0.81 | 48.07 | 1.56 | 29.83 | 1.46 | 34.25 | 1.78 | 4.43 |
| 0.7 | 32.40 | 0.74 | 46.02 | 1.52 | 35.97 | 1.45 | 37.55 | 1.53 | 3.54 |
| 0.8 | 29.90 | 0.84 | 42.78 | 1.65 | 29.60 | 1.55 | 28.65 | 1.82 | 7.96 |
| 0.9 | 12.75 | 0.81 | 46.00 | 1.53 | 31.10 | 1.73 | 27.64 | 1.72 | 10.78 |

# I Overall algorithm

To improve clarity, we provided an overall algorithm as follow.

---
**Algorithm 1** DAA Training and Test-Time Discovery with STMR

---
**Require:** $\mathcal{D}_{\text{train}} = \{(x_i, y_i)\}_{i=1}^{N_{\text{train}}}$ with known classes $\mathcal{Y}_{\text{kn}}$; test stream $\mathcal{D}_{\text{test}}$ with unknown classes; frozen backbone $f$, trainable adapter $\text{DAA}_\theta$; warm-up epochs $E$, memory size $M$, confidence threshold $\gamma$.

**Ensure:** Predictions $\hat{y}$ for test samples, dynamically updated $\theta$ and prototypes.

    **Phase 1: Discrepancy-Amplifying Warm-up (Pre-Testing)**
1:  initialize $\theta$ randomly
2:  **for** epoch $= 1, \ldots, E$ **do**
3:     **for** mini-batch $B \subseteq \mathcal{D}_{\text{train}}$ **do**
4:         $r \leftarrow f(B)$                                   ▷ backbone features
5:         $\tilde{r} \leftarrow \text{mixup}(r) + \mathcal{N}(0, \Sigma)$             ▷ synthetic unknowns
6:         $\mathcal{L}_{\text{kn}} \leftarrow \text{MSE}(\text{DAA}_\theta(r),\, r)$              ▷ preserve known
7:         $\mathcal{L}_{\text{un}} \leftarrow \text{contrastive}(\text{DAA}_\theta(r),\, \text{DAA}_\theta(\tilde{r}))$    ▷ amplify discrepancy
8:         $\mathcal{L}_{\text{train}} \leftarrow \mathcal{L}_{\text{kn}} + \lambda \mathcal{L}_{\text{un}}$
9:         $\theta \leftarrow \theta - \eta \nabla_\theta \mathcal{L}_{\text{train}}$
10:    **end for**
11: **end for**
    **Phase 2: Test-Time Discovery with STMR**
12: $(\mathcal{S}_{\text{kn}}, \mathcal{F}_{\text{kn}}) \leftarrow \text{load-known-prototypes}(\mathcal{D}_{\text{train}})$
13: $\mathcal{S}_{\text{un}} \leftarrow \{\}$; $\mathcal{F}_{\text{un}} \leftarrow \{\}$; $\mathcal{Y}_{\text{seen}} \leftarrow \mathcal{Y}_{\text{kn}}$
14: **for** each test batch $B$ **do**
15:    $r \leftarrow f(B)$; $z \leftarrow \text{DAA}_\theta(r)$
16:    compute cosine similarity $P(c)$ between $z$ and all prototypes $\{p_c \mid c \in \mathcal{Y}_{\text{seen}}\}$
17:    $\hat{y} \leftarrow_c P(c)$
18:    **if** $\max P > \gamma$ **then**
19:       assign $\hat{y}$                                      ▷ known/seen class
20:    **else**
21:       $\hat{y} \leftarrow$ "new_unknown"; $\mathcal{Y}_{\text{seen}} \leftarrow \mathcal{Y}_{\text{seen}} \cup \{\hat{y}\}$
22:       initialize new prototype $p_{\hat{y}} \leftarrow \text{mean}(z)$
23:    **end if**
                                                 ▷ STMR memory renewal (every $T$ batches)
24:    **if** $\hat{y} \in \mathcal{Y}_{\text{un}}$ **and** $\text{batch\_id} \bmod T = 0$ **then**
25:       **for** $x \in \mathcal{S}_{\text{un}}[\hat{y}]$ **do**
26:          $z_{\text{renew}} \leftarrow \text{DAA}_\theta(f(x))$
27:          **if** $_c P(z_{\text{renew}}) \in \mathcal{Y}_{\text{kn}}$ **then**
28:             discard $x$                      ▷ remove mis-classified known
29:          **else**
30:             $\mathcal{F}_{\text{un}}[\hat{y}].\,\text{enqueue}(z_{\text{renew}})$
31:          **end if**
32:       **end for**
33:    **end if**
34:    FIFO update:
35:    $\mathcal{S}_{\text{un}}[\hat{y}].\,\text{enqueue}(B)$; $\mathcal{F}_{\text{un}}[\hat{y}].\,\text{enqueue}(z)$
36:    $\mathcal{L}_{\text{test}} \leftarrow \text{compute-loss}(z, \hat{y}, \mathcal{S}_{\text{un}}, \mathcal{F}_{\text{un}})$              ▷ Eq. (6)
37:    $\theta \leftarrow \theta - \eta \nabla_\theta \mathcal{L}_{\text{test}}$                      ▷ self-supervised update
38: **end for**

---

