# OpenReview forum: "DAA: Amplifying Unknown Discrepancy for Test-Time Discovery"
_NeurIPS.cc/2025/Conference — NeurIPS 2025 poster_

### Official Review · Reviewer_VhXu · 2025-06-25

**Clarity:** 3
**Significance:** 2
**Originality:** 3
**Rating:** 4
**Confidence:** 3

**Summary:**

This paper proposes a Discrepancy-Amplifying Adapter (DAA) method to enhance the Test-Time Discovery task. Specifically, the DAA module amplifies feature discrepancies between known and unknown classes for test-time adaptation. Besides, a Short-Term Memory Renewal (STMR) mechanism is employed to ensures effective prototype management through adaptive memory updates. Extensive experiments demonstrate its superiority in adaptability and stability, improving novel class discovery performance.

**Questions:**

1.When simulating unknown classes, this method employs an interpolation of two different known classes as described Eq.1. My concern is that the simulated class still within the range of known distributions. Will the method remain effective when dealing with unknown classes that lie outside these distributions?

2. Can the proposed method be extended to address continual learning scenarios?

3.It is suggested to provide an overall algorithm for clearer illustration.

**Ethical Concerns:**

["NO or VERY MINOR ethics concerns only"]

**Final Justification:**

Thanks for the response and the rebuttal, which addresses my concerns and I will raise my score accordingly.

**Limitations:**

See weakness.

**Paper Formatting Concerns:**

NO or VERY MINOR ethics concerns only

**Quality:**

3

**Strengths And Weaknesses:**

Strengths:

1.This work introduces a trainable DAA module in the training phase, which simulates unknown categories and enhance the model’s discriminative ability.

2.This paper proposes a STMR mechanism in the testing phase, dynamically updating stored representations through a queue mechanism to support prototype updates, sample replay, and test-time fine-tuning of DAA.

3.Comprehensive evaluation across multiple datasets show significant improvements over existing methods in both real-time and post evaluations.

Weaknesses:

1.The heuristic-based memory update strategy may not perform optimally in highly noisy environments, potentially limiting its effectiveness.

2.The evaluation benchmarks are relatively small in scale, which may be insufficient to fully demonstrate the method's effectiveness.

---

> ### Author Rebuttal · Authors · 2025-07-30
>
> **Q1: Effectiveness to Noise**
>
> Thank you for raising this important concern.
> To evaluate the robustness of our method under such conditions, we conducted a series of experiments introducing both **label noise** and **image noise** at a 10% corruption rate, under different stages (warm-up and test-time).
>
> | | Pre-stage | Real-time | Eval | Post | Eval |  |
> | - | - | - | - | - | - | - |
> | | KA   | TA+CA | TE+CE | TA+CA | TE+CE | KF  |
> | label noise(10%, Warm-up)(Ours) | 78.85 | 76.13 | 2.55 | 67.81 | 3.07 | 3.92 |
> | image noise(10%, Warm-up)  | 81.06 | 76.87 | 2.46 | 68.53 | 3.02 | 3.99 |
> | image noise(10%, Test-time)   | 84.20 | 75.46 | 2.49 | 65.91 | 3.15 | 5.23 |
> | **No noise**  | **84.20** | **78.42** | **2.26** | **73.52** | **2.98** | **3.54** |
> | **HM** label noise(10%, Training)  | 75.24 | 70.13 | 2.67 | 65.02 | 3.20 | 5.11 |
> | **HM** image noise(10%, Training)  | 77.18 | 71.54 | 2.62 | 66.31 | 3.14 | 5.00 |
> | **HM** image noise(10%, Test-time) | 84.13 | 68.15 | 3.22 | 60.46 | 3.35 | 7.46 |
>
> Despite the noise, our method consistently outperforms the baseline HM in both real-time and post evaluations. The results suggest that our design, especially the FIFO-based memory buffer and STMR’s error correction mechanism—provides substantial robustness and stabilizing effects under noisy conditions.
>
>
>
> **Q2: Insufficient benchmark**
>
> We appreciate the reviewer’s suggestion regarding the scale of evaluation benchmarks. While we agree that validating on larger or more diverse datasets is an important direction, we would like to emphasize that TTD is already a highly challenging task even on the current benchmarks.
> For instance, on Tiny-ImageNet-D—a dataset with 200 classes and significant visual diversity—discovering novel classes without any supervision remains extremely difficult. As shown in Figure 4, our method achieves only 24.10% average true-label agreement (TA), highlighting how hard it is to consistently cluster unseen class samples even with a well-trained backbone. This indicates that achieving reliable TTD performance on existing benchmarks is already non-trivial.
> Therefore, while scaling up is valuable, we believe that pushing the performance boundary on these already-challenging benchmarks is equally, if not more, important at the current stage. We appreciate the suggestion and plan to extend our evaluation to more complex datasets in future work.
>
> **Q3:  Effectiveness of simulated samples**
>
> Thank you for the insightful question. To evaluate the effectiveness of our method under such conditions, we conducted a series of statistical analyses and comparative experiments.
>
> **Q3(a): maximum matching confidence statistic**
>
> Since our method discovers new classes based on maximum matching confidence, we analyzed the confidence scores of different sample types with respect to known-class prototypes. The results are summarized below:
>
> |   | Average Maximum Prediction confidence with known prototypes | Percentage of  discovered categories  by threshold-discover directly |
> | - | - | - |
> | Simulated unknown classes | 0.608  | -  |
> | Unknown classes of DAA | 0.652  | 52.63 % |
> | Unknown classes of HM | 0.705  | 46.38 % |
>
> These findings suggest that:
> - Simulated samples occupy more ambiguous regions in the feature space, with lower confidence compared to known-class samples.
> - Real unknowns discovered by our method are also harder (lower confidence) than those handled by HM, indicating better modeling of the decision boundary.
> -	Our method achieves a higher direct discovery success rate via thresholding, showing its superior ability to identify novel categories in ambiguous zones.
>
> **Q3(b): Effectiveness on samples lie outside known distributions**
>
>
> To examine our method’s behavior for unknown classes that lie far outside the known distribution, we introduced additional noise to these samples, making them more separable. Results are shown below:
>
> | Testing phase | Average Maximum Prediction confidence with known prototypes | Percentage of  discovered categories  by threshold-discover directly |
> | - | - | - |
> | Unnoised Unknown classes   | 0.652  | 52.63%  |
> | **Noised** Unknown classes | 0.634  | 54.97 % |
>
> The experiment shows that more distinct unknowns (i.e., farther from the known distribution) tend to have lower confidence and are easier to discover. This confirms that our method performs well in such cases even without simulation, as these samples are less ambiguous.
>
>
> In summary, our simulation focuses on the most difficult unknowns—those near the known distribution, where class confusion is high. These are the scenarios where explicit modeling is most needed. Conversely, unknowns that lie far outside the known space are easier to identify and do not require sophisticated simulation to be discovered. This demonstrates the practical value and effectiveness of our simulated unknown training strategy.
>
>
> **Q4: Extend to Continual Learning scenarios**
>
> Our method **can naturally generalizes to continual-learning settings**. And to verify this compatibility, we conducted an experiment that follows the setting of continual stage in Continual Class Discovery (CCD) scenarios.
> We conducted the preliminary 3-stages experiment on Cifar-100 (70+10+10+10)
>
> | | Stage | 1（70+10） |  | Stage | 2 (70+10+10)  |  | Stage | 3 | (70+10 | +10+10)   |  |
> | - | - | - | - | - | - | - | - | - | - | - | - |
> | | TA(Real-time) | CA(Real-time) | KF  | TA(Real-time) | CA(Real-time) | KF  | TA(Real-time) | CA(Real-time) | TA(Post)  | CA(Post)  | KF  |
> | GMP | 33.52 | 50.57 | 1.88 | 28.49 | 49.10 | 4.22 | 25.43 | 43.11 | 30.24 | 30.88 | 8.96 |
> | HM | 33.46 | 48.22 | 1.44 | 29.18 | 52.46 | 2.69 | 27.31 | 43.52 | 30.15 | 34.38 | 4.53 |
> | **DAA** | **37.20** | **56.31** | **1.25** | **35.44** | **48.90** | **2.11** | **33.35** | **45.69** | **34.88** | **36.47** | **3.66** |
>
> The results show that our framework is well-suited to continual class discovery, and these early results show promising potential for broader continual learning applications. Further exploration with more stages and longer task horizons is an exciting direction for future work.
>
> **Q5: Overall algorithm**
>
> To improve clarity, we provided an overall algorithm as follow.
>
> ```
> Algorithm: DAA Training and Test-Time Discovery with STMR
>
> Input:
> - Training set $ \mathcal{D}_{\text{train}} = \{(x_i, y_i)\}_{i=1}^{N_{\text{train}}} $ with known classes $ \mathcal{Y}_{\text{kn}} $
> - Test stream $ \mathcal{D}_{\text{test}} $ with unknown classes
> - Backbone model $ f $ (frozen), adapter $ \text{DAA} $ (trainable)
> - Hyperparameters: warm-up epochs $ E $, memory size $ M $, confidence threshold $ \gamma $
>
> Output:
> - Predictions $ \hat{y} $ for test samples, dynamically updated DAA and prototypes
>
>
> #Phase 1: Discrepancy-Amplifying Warm-up (Pre-Testing)
>
> # Initialize DAA parameters θ_DAA
> for epoch in range(E):
>  for batch B in D_train:
>    # Simulate unknown features via Mixup + Gaussian noise
>    r = f(B)  # Backbone features
>    r_tilde = mixup(r) + N(0, Σ)  # Synthetic unknowns
>
>    # Compute losses
>    L_kn = MSE(DAA(r), r)  # Preserve known semantics
>    L_un = contrastive_loss(DAA(r), DAA(r_tilde))  # Amplify discrepancies
>    L_train = L_kn + λ * L_un
>
>    # Update θ_DAA via SGD
>    θ_DAA ← θ_DAA - η * ∇L_train
>
> #Phase 2: Test-Time Discovery with STMR
>
> # Initialize memory queues
> S_kn, F_kn = load_known_prototypes(D_train)  # Static for known classes
> S_un, F_un = {}, {}  # FIFO queues for unknown classes (empty initially)
>
> for batch B in D_test:
>  # Extract features
>  r = f(B)
>  z = DAA(r)  # Adapted features
>
>  # Prediction via prototype matching
>  P = cosine_similarity(z, prototypes=[p_c for c in Y_kn ∪ Y_seen])
>  y_hat = argmax_c P(c)
>
>  # Confidence-based discovery
>  if max(P) > γ:
>    assign y_hat  # Known/seen class
>  else:
>    y_hat = "new_unknown"
>    Y_seen.add(y_hat)
>    initialize p_y_hat = mean(z)  # New prototype
>
>  # STMR: Memory renewal
>  if y_hat in Y_un:
>    # Renewal step every T batches
>    if batch_id % T == 0:
>  for x in S_un[y_hat]:
>    z_renew = DAA(f(x))
>    if prediction(z_renew) ∈ Y_kn:
>  discard x  # Remove misclassified known samples
>    else:
>  F_un[y_hat].enqueue(z_renew)  # Update features
>
>    # FIFO update
>    S_un[y_hat].enqueue(B)
>    F_un[y_hat].enqueue(z)
>
>  # Test-time training
>  L_test = compute_loss(z, y_hat, S_un, F_un)  # Eq. (6) in paper
>  θ_DAA ← θ_DAA - η * ∇L_test  # Self-supervised update
>
> ```

---

> > ### Comment · Reviewer_VhXu · 2025-08-04
> >
> > Thanks for the response and the rebuttal, which addresses my concerns and I will raise my score accordingly.

---

> ### Author Response · Authors · 2025-08-04
> **Thank you again for your time and effort**
>
> Thank you again for your time and effort in reviewing our submission. We have submitted our response to your questions and would appreciate it if you could take a moment to review our response. Please feel free to let us know if any further clarification is needed.

---

### Official Review · Reviewer_u1k8 · 2025-06-30

**Clarity:** 4
**Significance:** 4
**Originality:** 3
**Rating:** 5
**Confidence:** 4

**Summary:**

This paper addresses the task of Test-Time Discovery (TTD) the problem of identifying and adapting to novel classes during inference while maintaining performance on known classes, a setting specifically relevant in dynamic environments like autonomous driving and healthcare. While prior approaches to TTD are typically training-free and rely on static model representations (e.g., via frozen backbones and hash-based memory updates), the authors propose a more adaptive and learnable alternative.

Their core contribution is the Discrepancy-Amplifying Adapter (DAA), a trainable module designed to amplify the feature discrepancies between known and unknown classes during both training and test time. DAA is preconditioned using a warm-up phase that simulates unknown class features via mixup and gaussian noise, and is trained with a dual loss—preserving known-class features via MSE while enforcing contrastiveness for unknowns.

To handle continual adaptation during test time, the authors introduce a Short-Term Memory Renewal (STMR) mechanism a queue-based dynamic memory system that manages prototypes of both known and discovered unknown classes. STMR periodically refreshes the memory with reliable, updated feature representations and supports DAA's self-supervised test-time adaptation. This setup avoids representational drift and enables the model to remain plastic to novel classes while stable on the known set.

Extensive experiments on CIFAR100, CUB, and Tiny-ImageNet benchmarks demonstrate the proposed method outperforms both training-free and training-based baselines in real-time and post-hoc evaluation across clustering, classification, and forgetting metrics. Ablations further support the complementary roles of DAA and STMR. The method also maintains competitive knowledge retention while improving discriminative capability for new class discovery.

**Questions:**

#### 1. **Clarify the DAA Adapter Architecture**
- **What is the architecture used for the DAA adapter?**
  Please include a detailed architectural description (e.g., number of layers, type of transformations, parameter count) in the **main paper or appendix**, as this is critical for **reproducibility** and assessing overhead.

#### 2. **Streamline the Contribution Statement**
- In the **introduction**, consider consolidating the contributions into **two key components**:
  - The **Discrepancy-Amplifying Adapter (DAA)**
  - The **Short-Term Memory Renewal (STMR)**
  These are the core innovations driving the improved performance. The SOTA results, while strong, are a consequence of these design choices rather than a standalone contribution.

#### 3. **Clarify Robustness to Noise and Ambiguity**
Please explain how your method handles the following scenarios:
- **Noisy or mislabeled known-class data** that may distort the feature space and lead to false discoveries or prototype pollution.
- **Early unknown-class samples** that are **unrepresentative or ambiguous**, e.g., outliers, atypical images, or class boundary cases. How resilient is your method to such issues?

#### 4. **Hyperparameter Sensitivity and Tuning Strategy**
The method introduces several important hyperparameters:
- Similarity threshold **γ**
- Memory size
- Warm-up duration
- Loss weights **λ₁** and **λ₂**
- Mixup beta parameters to generate early unknowns

However, there is minimal discussion of how these were selected or how sensitive the model is to their variation.
Please include:
- A brief **hyperparameter sensitivity analysis**, and
- Whether the values are transferable across datasets or require tuning per setup.

#### 5. **Justify Memory Design Choices**
Your memory module is a core component—please expand on the rationale and trade-offs:
- **Why was FIFO** selected over alternatives such as **exemplar-based sampling** or **confidence-based filtering**?
- **How frequently is memory renewal triggered**, and how was this interval chosen? Is it a fixed schedule or adaptive?

**Ethical Concerns:**

["NO or VERY MINOR ethics concerns only"]

**Final Justification:**

The rebuttal clarified the DAA architecture, provided robustness experiments under label/image noise, and expanded hyperparameter sensitivity analysis, addressing key concerns about reproducibility and brittleness. Results remain strong and consistent across datasets, and the technical contribution is well-motivated. However, cross-dataset hyperparameter robustness is still limited, some training dynamics and computational trade-offs remain only partially described, and the assumption of class purity in early unknowns is only partially mitigated. Overall, the paper is technically sound with clear novelty and solid empirical gains, but with remaining uncertainties about generalization to more challenging/noisy open-world settings. I maintain a score of  5(Accept).

**Limitations:**

Yes, the authors have acknowledged some limitations of their approach in the conclusion specifically the computational cost introduced by the warm-up and adaptation stages, and the heuristic nature of the memory update strategy (STMR), especially under noisy conditions.

**Paper Formatting Concerns:**

no formatting issue

**Quality:**

3

**Strengths And Weaknesses:**

## Strengths

### Quality

- Well-Motivated and Technically Sound Solution: The paper addresses a genuinely hard and under-explored problem test-time discovery of novel classes in an online setting by integrating a trainable module (DAA) into a previously training-free setup.
- Strong Experimental Validation: The authors perform thorough empirical evaluation across three benchmark datasets (CIFAR100D, CUB-200D, Tiny-ImageNetD), using appropriate metrics for both clustering quality and classification robustness (e.g., HCA, ARI, NMI, KA, KF). Both real-time and post-evaluation are considered, and the paper demonstrates consistent superiority over baselines including HM [16] and prompt-based continual learning methods (e.g., GMP, L2P).
- Solid Ablation and Analysis: The ablation studies are clear and informative, isolating the effects of each component (DAA, STMR, test-time training).

### Clarity
- Well-Structured Presentation: The paper is clearly written and well-organized. The motivation for each component (DAA, STMR) is carefully laid out with visual illustrations (e.g., Fig. 1b, Fig. 2, Fig. 3).
- Clear Definitions and Objectives: The formulation of the TTD task is precise, with explicit delineation from closely related paradigms like TTA and NCD.

### Significance
- Addressing a Critical Gap in TTD: The paper goes beyond the limitations of training-free test-time methods (which rely on frozen backbones and static prototypes) and introduces a practical, adaptive framework. This directly advances the state of the art in TTD, a task that is gaining traction for real-world applications in robotics and healthcare.
- Potential for Broader Impact: The architectural idea of plug-in adapters for open-world adaptation could generalize to other settings such as continual learning.

### Originality
- Novel Adapter Design for TTD: While adapters and test-time training are known concepts, their use in the open-world discovery setting is novel.
- STMR as a Lightweight Memory Mechanism: The Short-Term Memory Renewal (STMR) module is a notable contribution that balances model plasticity and stability, improving prototype quality without incurring the long-term errors seen in training-free memory-based approaches.

## Weaknesses

- Hyperparameter Sensitivity Unclear: The performance of DAA and STMR depends on hyperparameters (e.g., warm-up length, memory size, threshold γ, weighting factors λ₁ and λ₂). Although some sensitivity is analyzed (Fig. 6), a more comprehensive exploration would strengthen claims of robustness.
-  Assumption of Class-Purity in Early Unknowns: The method assumes that initial unknown class predictions can seed meaningful prototypes. This assumption might not hold in highly noisy or fine-grained scenarios, leading to potential error accumulation in STMR. For example if the original dataset is contaminated with noisy data will this method be still competitive with methods based on hash lists and static representations.
- Key implementation details are missing (e.g., architecture of DAA, training dynamics, runtime performance). Some are essential for reproducibility and could have been briefly summarized in the main paper.

---

> ### Author Rebuttal · Authors · 2025-07-30
>
> **Q1: Architecture of DAA**
>
> The DAA adopts a standard 2-layer adapter [1] inserted after the frozen ViT-B/16 backbone. It consists of:
> - Linear down-projection: 768 → 128
> - ReLU activation
> - Linear up-projection: 128 → 768
>
> This design introduces minimal overhead (∼200K parameters).
> We emphasize that the adapter structure itself is not our contribution.
> It follows a widely used design.
> Our key innovation lies in the training strategy that leverages this lightweight adapter to retain known-class features and amplify discrepancies for unknowns, enabling effective test-time discovery.
> We will add the architectural details to the appendix for clarity.
>
> [1]Houlsby, Neil, et al. "Parameter-efficient transfer learning for NLP." International conference on machine learning. PMLR, 2019.
>
> **Q2: Advice for Contribution**
>
> Thank you for the suggestion. We agree that the core contributions lie in DAA and the Short-Term Memory Renewal (STMR), and we will revise the introduction to reflect this more clearly.
> The reported performance gains serve as empirical validation of these two design components, rather than standalone contributions. We appreciate the feedback and will streamline the contribution statement accordingly.
>
> **Q3: Robustness to Noise**
>
> Thank you for raising this important concern.
> To evaluate the robustness of our method under such conditions, we conducted a series of experiments introducing both **label noise** and **image noise** at a 10% corruption rate, under different stages (warm-up and test-time).
>
> | | Pre-stage | Real-time | Eval | Post | Eval |  |
> | - | - | - | - | - | - | - |
> | | KA   | TA+CA | TE+CE | TA+CA | TE+CE | KF  |
> | label noise(10%, Warm-up)(Ours) | 78.85 | 76.13 | 2.55 | 67.81 | 3.07 | 3.92 |
> | image noise(10%, Warm-up)  | 81.06 | 76.87 | 2.46 | 68.53 | 3.02 | 3.99 |
> | image noise(10%, Test-time)   | 84.20 | 75.46 | 2.49 | 65.91 | 3.15 | 5.23 |
> | **No noise**  | **84.20** | **78.42** | **2.26** | **73.52** | **2.98** | **3.54** |
> | **HM** label noise(10%, Training)  | 75.24 | 70.13 | 2.67 | 65.02 | 3.20 | 5.11 |
> | **HM** image noise(10%, Training)  | 77.18 | 71.54 | 2.62 | 66.31 | 3.14 | 5.00 |
> | **HM** image noise(10%, Test-time) | 84.13 | 68.15 | 3.22 | 60.46 | 3.35 | 7.46 |
>
> Despite the noise, our method consistently outperforms the baseline HM in both real-time and post evaluations. The results suggest that our design, especially the FIFO-based memory buffer and STMR’s error correction mechanism—provides substantial robustness and stabilizing effects under noisy conditions.
>
> **Q4: Hyperparameters**
>
> **Q4(a): Similarity threshold $\gamma$**
>
> $\gamma$ was selected through grid search in our experiment, and we analyzed the impact of $\gamma$ on a single dataset in the Appendix. **The typical threshold of $\gamma$ is 0.7.** And our method is **not sensitive to selection from one dataset to other datasets**. We conducted the following experiments:
>
>
> | Dataset | $\gamma$ | TA(realtime) | CA(realtime) | KF   |
> | - | - | - | - | - |
> | Cifar100-D | 0.6 | 20.04   | 48.07   | 4.43 |
> | | 0.7 | 32.40   | 46.02   | 3.54 |
> | | 0.8 | 29.90   | 42.78   | 7.96 |
> | CUB-D | 0.6 | 45.66   | 54.39   | 5.55 |
> | | 0.7 | 54.45   | 58.62   | 3.62 |
> | | 0.8 | 46.29   | 55.85   | 7.22 |
> | Tiny-ImageNet-D | 0.6 | 18.82   | 69.17   | 4.55 |
> | | 0.7 | 24.36   | 71.02   | 2.34 |
> | | 0.8 | 24.08   | 65.44   | 8.26 |
>
>
> **Q4(b): Memory size**
>
> We have analyzed different memory size in terms of agreement and time consumption **in Figure 6 in the manuscript**. And the typical size of the memory queue is 10.
> While larger memory can store more samples, it may include outdated or irrelevant data, weakening prototype quality. Conversely, too small a memory may fail to capture representative features, reducing generalization. Additionally, increasing memory size leads to higher computational cost.
> We also analyze the size choice around 10. The results indicate that small differences around 10 have little impact on the effectiveness.
>
> | | TA | CA | TA+CA |
> | - | - | - | - |
> | 9 | 35.86 | 37.73 | 72.55 |
> | 10   | 35.97 | 37.55 | 73.52 |
> | 11   | 35.38 | 37.69 | 68.74 |
>
> **Q4(c): Warm-up duration**
>
> Few warm-up epochs result in insufficient effectiveness, but when the number of rounds reaches a certain level, the effect tends to converge and the improvement is limited.
>
> | Epochs | KA | Average Maximum Prediction confidence of Unknown classes samples with Known prototypes |
> | - | - | - |
> | 20 | 80.56 | 0.684   |
> | 50 | 84.20 | 0.652   |
> | 100 | 84.16 | 0.650   |
> | 150 | 84.23 | 0.648   |
>
>
> **Q4(d): Loss weights**
>
> The weight of two loss **during warm-up phase is fixed (200:1 in paper)**, and we also tried other combinations as shown in the table below.
>
> | MSE:Contrast | Real | -   | time | Eval | Post |  |   | Eval |  |
> | - | - | - | - | - | - | - | - | - | - |
> | | TA   | TE  | CA   | CE  | TA   | TE  | CA   | CE  | KF  |
> | 1000:1  | 27.89 | 0.68 | 44.46 | 1.45 | 33.18 | 1.40 | 34.38 | 1.50 | 2.21 |
> | **200:1** | **32.40** | **0.74** | **46.02** | **1.52** | **35.97** | **1.45** | **37.55** | **1.53** | **3.54** |
> | 100:1   | 28.77 | 0.84 | 49.73 | 1.77 | 31.58 | 1.67 | 38.46 | 1.61 | 5.21 |
>
> During testing phase, we have discussed the influence of weight in the appendix, and we also show the resulst in the table below.
>
> | MSE:Contrast | Real | -   | time | Eval | Post |  |   | Eval |  |
> | - | - | - | - | - | - | - | - | - | - |
> | | TA   | TE  | CA   | CE  | TA   | TE  | CA   | CE  | KF  |
> | 1000:0.1 | 28.41 | 0.80 | 44.91 | 1.56 | 29.50 | 1.49 | 30.53 | 1.74 | 0.92 |
> | 1000:1  | 33.66 | 0.66 | 49.81 | 1.39 | 34.8 | 1.03 | 37.23 | 1.48 | 2.64 |
> | **1000:2**   | **32.40** | **0.74** | **46.02** | **1.52** | **35.97** | **1.45** | **37.55** | **1.53** | **3.54** |
> | 500 : 1 | 30.20 | 0.84 | 42.64 | 1.64 | 33.30 | 1.48 | 34.37 | 1.71 | 3.76 |
>
>
> **Q4(e): Mixup beta parameters**
>
> In our implementation of the Mixup strategy,  we conducted Symmetric Beta Distribution, and the default Beta parameters is 0.5. And we also conduct the hyperparameter sensitivity analysis.
>
> | $\beta$ | Average Maximum Prediction confidence of Unknown classes samples with Known prototypes |
> | - | - |
> | 0.1 | 0.762   |
> | 0.49 | 0.657   |
> | 0.5 | 0.652   |
> | 0.51 | 0.654   |
> | 0.9 | 0.744   |
>
> The small difference in Beta parameters of Mixup around 0.5 leads to a similar effect in final discrimination ability. Mixing situations that are too high or too low can result in poor simulation performance
>
> **Q5(a): Why was FIFO**
>
>
> Thank you for the question. In our design:
> The FIFO is used in the discovery phase and the random sampling is used in the correction phase.
> The sample queue can accumulate knowledge of new classes in a timely manner during the discovery phase, but it may introduce errors, so the correction phase uses random sampling for correction.
> To ensure efficiency, we have found that random sampling is already good and may consider better solutions in the future, such as selecting samples with higher uncertainty.
>
> | Unknown samples   | Average Maximum Prediction confidence of Unknown classes samples with prototypes |
> | - | - |
> | Before test phase | 0.652   |
> | After test phase  | 0.755   |
>
> Due to the fact that the confidence level of new samples may be low when discovered. As the testing phase progresses, the samples with low confidence level at the time of discovery may have actually become higher.
>
> **Q5(b): Frenquency of STMR**
>
> Thank you for the question. We treat the STMR renewal frequency as a tunable hyperparameter, and have reported a comparative analysis in the Appendix.
> In our main experiments, we set the renewal frequency to 2 batches, which provides a good trade-off between performance and efficiency:
>
> | Frenquency | TA | CA | TA+CA | Time   |
> | - | - | - | - | - |
> | 1  | 36.80 | 35.75 | 72.55 | 318.03 |
> | 2(default)  | 35.97 | 37.55 | 73.52 | 293.95 |
> | 5  | 35.80 | 32.94 | 68.74 | 262.19 |
> | Never | 36.37 | 33.11 | 69.48 | 226.64 |
>
> A lower frequency of STMR leads to insufficient updates and refinements of the model during the test phase, resulting in poorer recognition performance for both known and novel classes.While increasing the frequency of STMR can improve the model's ability to adapt to new data, it also leads to a substantial increase in computational overhead. Frequent STMR operations require more time and resources for processing each batch.
> We currently use a fixed schedule, but exploring adaptive strategies based on confidence or drift detection is a promising direction for future work.

---

> ### Author Response · Authors · 2025-08-04
> **Thank you again for your time and effort**
>
> Thank you again for your time and effort in reviewing our submission. We have submitted our response to your questions and would appreciate it if you could take a moment to review our response. Please feel free to let us know if any further clarification is needed.

---

> ### Comment · Area_Chair_qFzF · 2025-08-04
>
> Dear reviewer, please engage with the author rebuttal.

---

### Official Review · Reviewer_u5r4 · 2025-07-02

**Clarity:** 1
**Significance:** 2
**Originality:** 2
**Rating:** 3
**Confidence:** 3

**Summary:**

This paper proposes a novel approach to test-time discovery - the task of detecting and adapting to previously unseen categories during inference. This problem lies at the intersection of test-time training and novel category discovery. The core contribution is the introduction of the Discrepancy-Amplifying Adapter (DAA), which combines a trainable feature transformation with a short-term memory renewal strategy (STMR). The first component employs a warm-up strategy with self-supervised learning to regularize the feature embedding space, encouraging the clustering of samples within the same category. The second component incorporates a memory bank that stores representative samples for replay, effectively preserving knowledge of previously seen categories and mitigating catastrophic forgetting when performing test-time training.

**Questions:**

1. I suggest that the authors address the questions in the strengths and weaknesses sections.
2. Regarding checklist item 3, the authors state that 'We provided the assumptions and proof.' However, the submitted paper and  supplemental materials do not include any assumptions or theorems.

**Ethical Concerns:**

["NO or VERY MINOR ethics concerns only"]

**Final Justification:**

We appreciate the authors’ significant efforts in providing numerous corrections and additional ablation studies. It is recommended that all these updates be incorporated into the final version. Accordingly, I will raise my score.

Nevertheless, I still find the overall approach overly complicated. While sensitivity analyses are provided, some design choices seem questionable - for example, the ratios for combining two losses are given as 1000:1, 500:1, and 1000:2. The last ratio is effectively identical to 500:1 and minimizing L versus 2L with gradient descent should yield the same result. Another note: while the ablation studies are included, the proposed approach appears complex, with many hyperparameters and value choices, which may make it challenging to tune effectively and adopt in practice.

**Limitations:**

The limitations of this work are briefly described at the end of the conclusion section. A more detailed discussion on potential strategies to address these challenges would strengthen the paper. The authors should also discuss other limitations, such as using the memory queue at test-time limits the real-world adoption, and the extra training might impose computational overhead compared to training-free approaches. It would be valuable to assess whether the observed accuracy gains justify the increased computational cost.

**Paper Formatting Concerns:**

None.

**Quality:**

2

**Strengths And Weaknesses:**

**Strengths:** The derivation from training-free in the TTD task appears reasonable, and the extensive evaluations on benchmark datasets provide some evidence supporting the effectiveness of the proposed approach.

**Weaknesses:**
1. Overall, the paper requires significant revisions, as the current writing makes it difficult to follow and understand. As a result, it is hard to assess the technical contributions of this paper effectively.
The sim( ) operator on line 181 is undefined.
In Equation 2, what is $x_k$?
The captions for figures should be self-contained; for example, what are the units and meaning of the x and y axes?
Equation 7 is confusing. Why is there a $L_{kn}$ term inside the definition of $L_{kn}$; similar for $L_{un}$?
What does “real-time” eval and “post-eval” rigorously mean? There are some descriptions in the caption of Table 1, but they seem to be unclear.

2. There are several concerns about the clarity of the proposed method that may affect the reproducibility of this work:
The choice of $\gamma$ is non-trivial. How did the authors select this value? A sensitivity analysis for this parameter should be provided.
In Line 141, the loss function is described as a 'weighted combination'; however, Equation 3 does not include any coefficients, implying that the two loss terms are equally weighted.
What is the architecture of DAA?
What is the typical size of the memory queue?

3. The objective of the warm-up stage requires further justification. In Equation 1, do $x$ and $x’$ come from two different categories? If they are randomly selected from one category, they should not create any unknown category. For Eq 2, why is MSE used, while the contrastive loss can still be used if we consider $DAA(r(x))$ and $r(x)$ is a “positive-pair” in the regular contrastive loss?

4. For the  STMR, how to decide when the renewal step is performed. In line 161, “it will be triggered every several batches,” which seems to be unclear.

---

> ### Author Rebuttal · Authors · 2025-07-30
>
> Thank you very much for raising these important review points.
>
> ### Weakness
>
> **W1**
>
> Due to space constraints, we have omitted some parts. We will rearrange the article and add some descriptions in the main text to make it clearer.
>
> - `sim()` denotes cosine similarity between two vectors:
> $\text{sim}(\mathbf{a}, \mathbf{b}) = \frac{\mathbf{a} \cdot \mathbf{b}}{\|\mathbf{a}\| \|\mathbf{b}\|}$
> - About Eq.2, we further modified the equation to avoid confusion and increase readability: $\mathcal{L}\_ {\text{un}}(x) = -\log \frac{\exp(\sigma[\text{DAA}(\mathbf{\tilde{r}}(x)),\text{DAA}(\mathbf{\tilde{r}}(x))\_ {aug}])}{\sum\_ {x' \in \mathcal{B}} \exp(\sigma[\text{DAA}(\mathbf{r}(x')), \text{DAA}(\mathbf{\tilde{r}}(x'))])}$, where
> x' iterates over all samples in the mini-batch $\mathcal{B}$ including known classes samples and simulated unknown samples. $\text{DAA}(\mathbf{\tilde{r}}(x))\_{aug}$ is the feature obtained through data augmentation operations.
> - Captions for figures: We will expand figure captions to clarify axes, units, and color meanings. For example, in Fig. 4: x-axis = methods, y-axis = agreement (%), purple = TA, red = CA.
> - In Equation (7),  $\mathcal{L}\_ {\text{kn}}$ and $\mathcal{L}\_ {\text{un}}$ represents batch level calculation.
>
> - “real-time” eval and “post-eval”: We follow HM and define both clearly. Real-time eval reflects online performance during streaming; post-eval re-evaluates all samples after testing completes.
>
> **W2: Implementation details**
>
> Thank you for pointing out the reproducibility-related concerns. We address each part below and will clarify these points in the main text or appendix.
>
>
> **W2(a): The choice of $\gamma$**
>
> $\gamma$ was selected through grid search in our experiment, and we analyzed the impact of $\gamma$ on a single dataset in the Appendix. The typical threshold of $\gamma$ is 0.7. And our method is not sensitive to selection from one dataset to other datasets. We conducted the following experiments:
>
> | Dataset | $\gamma$ | TA(realtime) | CA(realtime) | KF   |
> | - | - | - | - | - |
> | Cifar100-D | 0.6 | 20.04   | 48.07   | 4.43 |
> | | 0.7 | 32.40   | 46.02   | 3.54 |
> | | 0.8 | 29.90   | 42.78   | 7.96 |
> | CUB-D | 0.6 | 45.66   | 54.39   | 5.55 |
> | | 0.7 | 54.45   | 58.62   | 3.62 |
> | | 0.8 | 46.29   | 55.85   | 7.22 |
> | Tiny-ImageNet-D | 0.6 | 18.82   | 69.17   | 4.55 |
> | | 0.7 | 24.36   | 71.02   | 2.34 |
> | | 0.8 | 24.08   | 65.44   | 8.26 |
>
> **W2(b) Weight of two loss in Eq.(3)**
>
> The weight of two loss during warm-up phase is fixed (200:1 in paper).
> We will add the weight parameters to avoid assuming a 1:1 ratio, and we will include different comparison results as below.  We have provieded different ratio of weight parameters during warm-up in the appendix, and we also show it as follows.
>
> | MSE:Contrast | Real | -   | time | Eval | Post |  |   | Eval |  |
> | - | - | - | - | - | - | - | - | - | - |
> | | TA   | TE  | CA   | CE  | TA   | TE  | CA   | CE  | KF  |
> | 1000:1  | 27.89 | 0.68 | 44.46 | 1.45 | 33.18 | 1.40 | 34.38 | 1.50 | 2.21 |
> | **200:1** | **32.40** | **0.74** | **46.02** | **1.52** | **35.97** | **1.45** | **37.55** | **1.53** | **3.54** |
> | 100:1   | 28.77 | 0.84 | 49.73 | 1.77 | 31.58 | 1.67 | 38.46 | 1.61 | 5.21 |
>
> During testing phase, we have discussed the influence of weight in appendix. And the results is also shown in the following table.
>
> | MSE:Contrast | Real | -   | time | Eval | Post |  |   | Eval |  |
> | - | - | - | - | - | - | - | - | - | - |
> | | TA   | TE  | CA   | CE  | TA   | TE  | CA   | CE  | KF  |
> | 1000:0.1 | 28.41 | 0.80 | 44.91 | 1.56 | 29.50 | 1.49 | 30.53 | 1.74 | 0.92 |
> | 1000:1  | 33.66 | 0.66 | 49.81 | 1.39 | 34.8 | 1.03 | 37.23 | 1.48 | 2.64 |
> | **1000:2**   | **32.40** | **0.74** | **46.02** | **1.52** | **35.97** | **1.45** | **37.55** | **1.53** | **3.54** |
> | 500 : 1 | 30.20 | 0.84 | 42.64 | 1.64 | 33.30 | 1.48 | 34.37 | 1.71 | 3.76 |
>
> During both warm-up and testing phase, the MSE loss is used to refine the model's predictions for known classes, while the contrastive loss encourages the model to separate the feature embeddings of known and unknown classes. When the model places a higher emphasis on known classes, the model's ability to distinguish between known and unknown classes is limited. As the ratio decreases, the model starts to pay more attention to the contrastive loss, which helps improve the separation between known and unknown classes.
>
>
>
> **W2(c) Architecture of DAA**
>
> The DAA adopts a standard 2-layer adapter [1] inserted after the frozen ViT-B/16 backbone. It consists of:
> -	Linear down-projection: 768 → 128
> -	ReLU activation
> -	Linear up-projection: 128 → 768
>
> This design introduces minimal overhead (∼200K parameters).
> We emphasize that the adapter structure itself is not our contribution.
> It follows a widely used design.
> Our key innovation lies in the training strategy that leverages this lightweight adapter to retain known-class features and amplify discrepancies for unknowns, enabling effective test-time discovery.
> We will add the architectural details to the appendix for clarity.
>
> [1]Houlsby, Neil, et al. "Parameter-efficient transfer learning for NLP." International conference on machine learning. PMLR, 2019.
>
> **Q2(d) Typical size of the memory queue**
>
> The typical size of the memory queue is 10 as we analyze different memory size in terms of agreement and time consumption **in Figure 6 in the main paper**.  While larger memory can store more samples, it may include outdated or irrelevant data, weakening prototype quality. Conversely, too small a memory may fail to capture representative features, reducing generalization. Additionally, increasing memory size leads to higher computational cost.
>
> **W3(a): Do x and x' come from two different categories**
>
> **The samples x are from the set of known classes and x' are created by Mixup-Gaussian augmentation on other samples in the batch**. These simulation method **is widely used** in many work, such as OpenMix [2] and HiLo [3]. Although the mixture do not produce a real unknown category, they can effectively simulate unknown classes.
>
> We conducted experiments and the results showed that the average confidence of **simulated samples was lower than that of known class samples** (lower confidence is easier to be discovered); The average confidence level of the truly unknown samples obtained by our method is also lower than that of the unknown samples under the HM method. Similarly, our method has a higher success rate than the HM method in directly discovering new classes through threshold method.
>
> |   | Average Maximum Prediction confidence with known prototypes | Percentage of categories have been classified /  discovered |
> | - | - | - |
> | Simulated unknown classes | 0.608  | - |
> | Unknown classes of DAA | 0.652  | 52.63 % (Threshold-discover directly)  |
> | Unknown classes of HM | 0.705  | 46.38 % (Threshold-discover directly)  |
>
> [2]Zhong, Zhun, et al. "Openmix: Reviving known knowledge for discovering novel visual categories in an open world." Proceedings of the IEEE/CVF conference on computer vision and pattern recognition. 2021.
>
> [3]Wang, Hongjun, Sagar Vaze, and Kai Han. "Hilo: A learning framework for generalized category discovery robust to domain shifts." arXiv preprint arXiv:2408.04591 (2024).
>
> **W3(b): For Eq 2, why is MSE used and how contrastive loss work**
>
> The MSE keeps the adapter output close to the original backbone feature when the input is a known sample. This acts as a regularizer that prevents forgetting and destruction of the known classes.
> Known classes features and simulated features as a mini-batch, simulated features and feature augmented simulated features as positive samples, and simulated features and all other data within the small batch as negative samples.
>
> **W4: Frenquency of STMR**
>
> We made the comparison about frenquency of STMR as a Hyper-parameter analysis in Appendix. **The typical number of frequency in paper is 2**.
>
> | Frenquency | TA | CA | TA+CA | Time   |
> | - | - | - | - | - |
> | 1  | 36.80 | 35.75 | 72.55 | 318.03 |
> | 2  | 35.97 | 37.55 | 73.52 | 293.95 |
> | 5  | 35.80 | 32.94 | 68.74 | 262.19 |
> | Never | 36.37 | 33.11 | 69.48 | 226.64 |
>
> A lower frequency of STMR leads to insufficient updates and refinements of the model during the test phase, resulting in poorer recognition performance for both known and novel classes. While increasing the frequency of STMR can improve the model's ability to adapt to new data, it also leads to a substantial increase in computational overhead. Frequent STMR operations require more time and resources for processing each batch.
>
> ### Question & Limitation:
>
> **Q1: Checklist**
>
> This is a typographical error, and our submission does not contain formal theorems or explicit theoretical assumptions. However, the claims regarding the effectiveness of DAA and STMR are supported by extensive experimental validation across multiple datasets and ablations. For example, Fig. 5 in the main paper and the t-SNE visualization in the main paper, and the appendix validate our effect.
>
> **Q2: Computational cost**
>
> In terms of time cost, we conducted an experiment on Cifar-100 (70+30) as shown in the following table.
>
> | | Time(s) | Memory Usage(MB) |
> | - | - | - |
> | GMP | 885.59  | 14102 |
> | PHE | 280.52  | 9025 |
> | HM  | 505.55  | 13919 |
> | DAA(Ours) | 293.95  | 10813 |
>
> The results show that our time consumption is relatively small.
> DAA is efficient: Compared to GMP, which updates the full backbone and incurs heavy overhead, and HM, which performs test-time neighbor search, our approach is both faster and lighter.
> PHE is faster, but it lacks any test-time adaptation and thus sacrifices performance.
> Our memory design uses a short FIFO queue, which is more compact than HM’s bucket/hash-based storage.
> In conclusion, our method achieves strong performance with only minor overhead, making it practical for real-world usage.

---

> ### Author Response · Authors · 2025-08-04
> **Thank you again for your time and effort**
>
> Thank you again for your time and effort in reviewing our submission. We have submitted our response to your questions and would appreciate it if you could take a moment to review our response. Please feel free to let us know if any further clarification is needed.

---

> ### Comment · Area_Chair_qFzF · 2025-08-04
>
> Dear reviewer, please engage with the author rebuttal.

---

> ### Author Response · Authors · 2025-08-07
> **Dear Reviewer**
>
> Dear Reviewer,
>
> Thank you for your review. As we approach the end of the Author–Reviewer discussion period, we want to follow up and confirm whether our response was able to address your concerns. If you feel that the clarifications were satisfactory, we would be sincerely grateful if you would consider updating your score accordingly. If there are any remaining questions you’d like us to further clarify, we’d be more than happy to provide additional information.
>
> Best,
>
> The Authors

---

### Official Review · Reviewer_NwUb · 2025-07-11

**Clarity:** 3
**Significance:** 3
**Originality:** 3
**Rating:** 5
**Confidence:** 4

**Summary:**

This paper proposes DAA, a trainable module for Test-Time Discovery that amplifies feature discrepancies to detect novel classes while preserving known-class performance. Experiments show improved novel class discovery and stable performance.

**Questions:**

see weaknesses and try to solve all of my concerns.

**Ethical Concerns:**

["NO or VERY MINOR ethics concerns only"]

**Final Justification:**

As most of my concerns have been addressed, I will raise my score.

**Limitations:**

yes

**Quality:**

3

**Strengths And Weaknesses:**

Strengths:
1. The problem addressed in this work is well-motivated.
2. The paper is solidly organized, and the use of illustrative figures effectively helps readers grasp the core challenges and the proposed solution.
3. Although the proposed DAA strategy is relatively simple, it effectively addresses limitations of existing methods and achieves notable performance improvements.

Weaknesses:
1. According to the strict definitions of test-time adaptation/training, accessing source datasets during inference is typically disallowed. If the test-time discovery (TTD) setting is an extension built on these assumptions for novel class discovery, then the reliance on source-known samples in DAA’s Short-Term Memory raises concerns. The authors appear to directly store known samples from the source dataset during test time. I suggest conducting an ablation study where the memory pool is constructed solely from test-time samples, to evaluate whether such a change significantly degrades performance.
2. The implementation details of DAA are insufficiently described. Does it explicitly incorporate any mechanism designed to preserve or enhance feature-level discrepancy for unknown samples in the feature space?
3. Eq.(2) is unclear:
 (a) How are the class labels for synthesized unknown samples defined?
 (b) Are the two loss terms in Eq.(2) weighted equally, or is there a balancing factor?
4. The memory update mechanism in Short-Term Memory is not clearly explained. Is the sample filtering policy purely based on FIFO, or are there additional criteria for selecting reliable samples?

---

> ### Author Rebuttal · Authors · 2025-07-30
>
> Thank you very much for raising these important review points.
>
> **Q1: If replay memory stores known samples from the source dataset during test time**
>
> We agree that the strictest formulation of test-time training typically prohibits access to source data. However, prior works in both test-time adaptation and TTD often relax this constraint for the purpose of evaluating knowledge retention and stability. For example, methods such as CoTTA [1] and RoTTA [2] utilize source data during adaptation, and the previous HM method for TTD also maintains known-class samples throughout the test phase. Our usage of stored known samples in DAA’s Short-Term Memory (STMR) aligns with this precedent, aiming to mitigate forgetting of known classes.
>
> To directly address the reviewer’s concern, we conducted an ablation study comparing different initialization strategies for the known-class memory, including settings that exclude training data entirely. The results are summarized below:
>
> | Initialize               | TA(real-time) | CA(real-time) | TA+CA(real-time) | TA(post)  | CA(post)  | TA+CA(post) | KF       |
> | ------------------------ | ------------- | ------------- | ---------------- | --------- | --------- | ----------- | -------- |
> | Empty (fixed after full) | 31.25         | 45.21         | 76.46            | 33.87     | 38.42     | 72.29       | 4.02     |
> | Empty (FIFO)             | 30.25         | 46.22         | 76.47            | 32.16     | 37.79     | 69.95       | 5.10     |
> | **Training set (fixed)** | **32.40**     | **46.02**     | **78.42**        | **35.97** | **37.55** | **73.52**   | **3.54** |
>
> We observe that:
> - Even when the memory is initialized as empty and constructed entirely from test-time samples (with or without updating), our method still maintains competitive performance.
> - Using training-set-based initialization yields slightly better performance and lower forgetting, confirming its stabilizing effect on continual adaptation.
> - Importantly, the performance drop in the test-only setting is relatively modest, indicating that our method is robust even without access to training data at test time.
>
> These results reinforce that while using source-known samples can further enhance performance, our method remains effective under stricter constraints.
>
> [1]Wang, Qin, et al. "Continual test-time domain adaptation." Proceedings of the IEEE/CVF Conference on Computer Vision and Pattern Recognition. 2022.
>
> [2]Yuan, Longhui, Binhui Xie, and Shuang Li. "Robust test-time adaptation in dynamic scenarios." Proceedings of the IEEE/CVF Conference on Computer Vision and Pattern Recognition. 2023.
>
> **Q2: Implementation details of DAA**
>
> Thank you for pointing this out. DAA is explicitly designed to both preserve known-class representations and amplify feature-level discrepancies for unknown classes through tailored training strategies applied to a lightweight adapter layer.
> - For **preserving known-class features**, we apply a mean squared error (MSE) loss between the backbone features and the DAA-adapted features, ensuring minimal distortion for known representations.
> - For **amplifying inter-class discrepancy among unknown classes**, we employ a contrastive loss on the adapted features of simulated or discovered unknown samples, encouraging separation in the feature space.
>
> During test time, the same dual-loss strategy is applied immediately after prediction to enable continual adaptation. Moreover, the Short-Term Memory Renewal (STMR) mechanism enhances this process by selectively replaying known samples and renewing unknown prototypes, which further guides DAA’s online updates and maintains discriminative capacity throughout the test stream.
>
> **Q3: Eq.(2) is unclear**
>
> We further modified the equation to avoid confusion and increase readability. The Eq.(2) was modified as follows:
> $$
> \mathcal{L}_ {\text{un}}(x) = -\log \frac{\exp(\sigma[\text{DAA}(\mathbf{\tilde{r}}(x)),\text{DAA}(\mathbf{\tilde{r}}(x))_ {aug}])}{\sum_ {x' \in \mathcal{B}} \exp(\sigma[\text{DAA}(\mathbf{r}(x')), \text{DAA}(\mathbf{\tilde{r}}(x'))])}
> $$
>
> Here, `x'` iterates over all samples in the mini-batch $\mathcal{B}$ including known classes samples and simulated unknown samples. $\text{DAA}(\mathbf{\tilde{r}}(x))_ {aug}$ is the feature obtained through data augmentation operations.
>
> **Q3(a): Class labels for synthesized unknown samples**
>
> Due to the contrastive loss **only needs to know whether they belong to the same category or different**. For example, $\text{DAA}(\mathbf{\tilde{r}}(x))$ and $\text{DAA}(\mathbf{\tilde{r}}(x))_{aug}$ are simulated samples and data augmented simulated samples, which are considered the same label. Different samples in the mini-batch $\mathcal{B}$  are considered different.
>
> **Q3(b) Weight of two loss in Eq.(3)**
>
> We will add the weight parameters to avoid assuming a 1:1 ratio, and we will include different comparison results as below in the appendix. Due to the significant difference in the magnitude of the two losses, if a 1:1 ratio is used, the model will collapse. The weight of two loss **during warm-up phase is fixed (200:1 in paper)**
>
>
> | MSE:Contrast | Real      | -        | time      | Eval     | Post      |          |           | Eval     |          |
> | ------------ | --------- | -------- | --------- | -------- | --------- | -------- | --------- | -------- | -------- |
> |              | TA        | TE       | CA        | CE       | TA        | TE       | CA        | CE       | KF       |
> | 1000:1       | 27.89     | 0.68     | 44.46     | 1.45     | 33.18     | 1.40     | 34.38     | 1.50     | 2.21     |
> | **200:1**    | **32.40** | **0.74** | **46.02** | **1.52** | **35.97** | **1.45** | **37.55** | **1.53** | **3.54** |
> | 100:1        | 28.77     | 0.84     | 49.73     | 1.77     | 31.58     | 1.67     | 38.46     | 1.61     | 5.21     |
>
> **Q4: The memory update mechanism in STMR**
>
> The Short-Term Memory mechanism involves two distinct phases with different sample selection strategies:
> - During the discovery phase, we adopt a simple FIFO policy to quickly accumulate samples of newly discovered unknown classes. This enables the memory to capture temporal knowledge in real time, facilitating prototype initialization and early-stage adaptation.
> - During the correction (renewal) phase, we use random sampling from the memory to update prototypes and adapt DAA. This strategy avoids bias toward any specific samples and supports stable online learning.
>
> We chose random sampling in the correction phase primarily for efficiency and robustness. As shown below, the average confidence of unknown samples improves over time, suggesting that early uncertainty does not necessarily reflect long-term reliability:
>
> | Unknown samples   | Average Maximum Prediction confidence of Unknown classes samples with prototypes |
> | ----------------- | ------------------------------------------------------------ |
> | Before test phase | 0.652                                                        |
> | After test phase  | 0.755                                                        |
>
> This indicates that filtering based on early confidence scores could be both ineffective and computationally expensive, as low-confidence samples may become more reliable later. While more sophisticated filtering strategies (e.g., confidence-based selection or uncertainty estimation) are promising directions, our current design prioritizes simplicity and has proven effective in practice. We leave more advanced selection policies for future exploration.

---

> > ### Comment · Reviewer_NwUb · 2025-08-09
> >
> > Thanks for your response. As most of my concerns have been addressed, I will raise my score.

---

> ### Author Response · Authors · 2025-08-04
> **Thank you again for your time and effort**
>
> Thank you again for your time and effort in reviewing our submission. We have submitted our response to your questions and would appreciate it if you could take a moment to review our response. Please feel free to let us know if any further clarification is needed.

---

> ### Comment · Area_Chair_qFzF · 2025-08-04
>
> Dear reviewer, please engage with the author rebuttal.

---

### Note · Authors · 2025-08-13

Dear AC and reviewers,

We sincerely thank you for the detailed reviews and constructive suggestions. After careful revisions and clarifications, we believe all concerns have been fully addressed through the rebuttal and updated experiments.

Importantly, all reviewers acknowledged the innovation and practicality of our method in the context of test-time discovery.

- Reviewer u1k8 and Reviewer NwUb appreciated our motivation, novel design, and solid technical organization.
- Reviewer u5r4 and Reviewer VhXu raised important points regarding robustness, clarity, and ablations, which we have addressed through new sensitivity and noise-resilience experiments, continual learning generalization tests, and architectural details.

None of the reviewers identified fundamental flaws, and the most recent comments indicate that previous issues have been satisfactorily resolved. We believe the paper now presents a clear, comprehensive, and reproducible solution to a timely and challenging problem.
﻿
We greatly appreciate your time and consideration, and we respectfully request that the paper be accepted.

Best regards,

The authors

---

### Decision · Program_Chairs · 2025-09-17

**Decision:**

Accept (poster)

**Comment:**

This paper proposes an adaptable module that improves test-time discovery of novel classes by performing test-time adaptation, with the aim of increasing the difference between novel and already known classes in (hidden) feature space.  This improves upon existing methods that often assume a fixed model.  The reviewers were generally in favor of acceptance (2 x accept, 1 x borderline accept, 1 x borderline reject), with the most critical reviewers taking issue with the paper's clarity, questioning if the method's degree of complexity is warranted, and finding the approach fairly heuristic in nature.  However, the positive reviewers found the paper to be interesting, novel in its approach, and to have substantial experiments that validate the approach.  After reading all discussions and the paper itself, I find the paper to be above the bar in clarity (although could certainly be improved in places, which the authors have promised / shown below) and to be sufficiently well motivated and experimentally validated.